# TAID: Temporally Adaptive Interpolated Distillation for Efficient Knowledge Transfer in Language Models

**Makoto Shing**[1]**, Kou Misaki**[1]**, Han Bao**[2]**, Sho Yokoi**[345]**, Takuya Akiba**[1]
[1]Sakana AI, [2]Kyoto University, [3]NINJAL, [4]Tohoku University, [5]RIKEN
{mkshing,kou.misaki,takiba}@sakana.ai,bao@i.kyoto-u.ac.jp,
yokoi@ninjal.ac.jp

## Abstract

Causal language models have demonstrated remarkable capabilities, but their size poses significant challenges for deployment in resource-constrained environments. Knowledge distillation, a widely-used technique for transferring knowledge from a large teacher model to a small student model, presents a promising approach for model compression. A significant remaining issue lies in the major differences between teacher and student models, namely the substantial capacity gap, mode averaging, and mode collapse, which pose barriers during distillation.s To address these issues, we introduce *Temporally Adaptive Interpolated Distillation (TAID)*, a novel knowledge distillation approach that dynamically interpolates student and teacher distributions through an adaptive intermediate distribution, gradually shifting from the student's initial distribution towards the teacher's distribution. We provide a theoretical analysis demonstrating TAID's ability to prevent mode collapse and empirically show its effectiveness in addressing the capacity gap while balancing mode averaging and mode collapse. Our comprehensive experiments demonstrate TAID's superior performance across various model sizes and architectures in both instruction tuning and pre-training scenarios. Furthermore, we showcase TAID's practical impact by developing two state-of-the-art compact foundation models: `TAID-LLM-1.5B` for language tasks and `TAID-VLM-2B` for vision-language tasks. These results demonstrate TAID's effectiveness in creating high-performing and efficient models, advancing the development of more accessible AI technologies.

## 1 Introduction

**Large language models are too large.** Causal language models (**LMs**) are increasingly becoming essential tools across various sectors (Malinka et al., 2023; Wu et al., 2023; Zhang et al., 2023a; He et al., 2024). Scaling data size, model size, and training steps has been the primary approach to improve LM performance (Kaplan et al., 2020; Hoffmann et al., 2022; OpenAI et al., 2024), leading to rapid advancements in both proprietary and open-source LMs (Touvron et al., 2023; Abdin et al., 2024; Yang et al., 2024). However, the success of large LMs creates challenges: they are too large for edge devices (Qu et al., 2024; Thawakar et al., 2024; Liu et al., 2024), have decoding times too long for real-time applications (Wan et al., 2023; Leviathan et al., 2023; Miao et al., 2024), and consume significant energy resources (Luccioni et al., 2023; Faiz et al., 2024). This paradox of scale hinders the widespread deployment and use of LMs despite their potential and high demand.

**Knowledge distillation offers a promising prescription.** One promising approach to developing compact yet high-performing models is knowledge distillation (**KD**) (Hinton et al., 2015). KD aims to transfer the knowledge, specifically the predicted distributions, from a well-trained, high-capacity teacher model to a more compact student model, often achieving better performance than small models trained solely (Buciluundefined et al., 2006; Ba & Caruana, 2014; Hinton et al., 2015). In the context of compressing large LMs, KD is becoming a mainstream approach, with many specialized KD methods actively being developed (Xu et al., 2024; Team et al., 2024; Muralidharan et al., 2024).

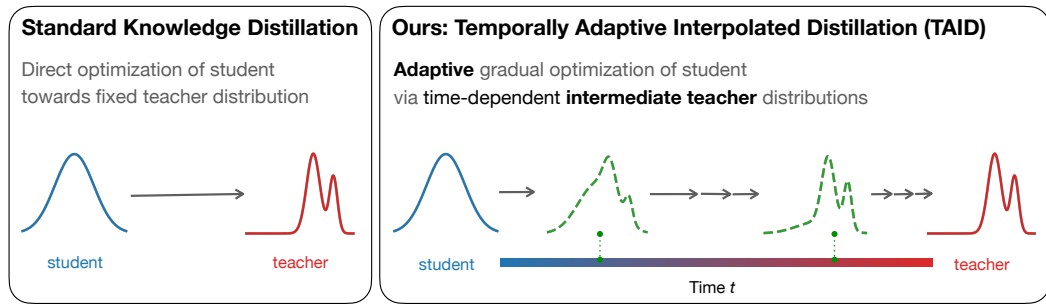

Figure 1: **Comparison of standard KD and TAID. (Left)** Standard KD methods typically employ direct optimization towards a fixed teacher distribution. **(Right)** TAID creates a dynamic bridge through adaptive, time-dependent *intermediate teacher* distributions (green dashed lines), enabling gradual optimization of the student. This approach facilitates a flexible transition from the student's initial distribution towards the teacher's distribution over time, effectively addressing the capacity gap and balancing knowledge transfer across varying model sizes.

**The formidable, unresolved challenge of teacher-student differences.** Nevertheless, KD is not a flawless method, and two significant issues remain, both stemming from the differences between teacher models and the student models.

(i) *Capacity gap* — the substantial capacity gap between a large teacher model and compact student model makes effective knowledge transfer more difficult (Mirzadeh et al., 2020; Cho & Hariharan, 2019; Zhang et al., 2023b). As LMs continue to grow in size and complexity, this capacity gap becomes increasingly pronounced, making it even more challenging to distill knowledge effectively. (ii) *Mode averaging and mode collapse* — due to the disparity in model capacity, KD methods often struggle with mode-averaging and mode-collapse issues, where student models either fail to oversmooth rich output distributions of a teacher model or become overly focused on specific modes (Wen et al., 2023; Gu et al., 2024; Agarwal et al., 2024).

**A new method to overcome the teacher-student difference.** To overcome the fundamental issue of differences between teacher and student models, we introduce Temporally Adaptive Interpolated Distillation (**TAID**), a new approach to KD for LMs. TAID reduces the gap between teacher and student model throughout the training process by dynamically introducing an *intermediate teacher* that interpolates teacher and student models to provide a target distribution with a modest capability (see Figure 1). This simple technique allows for learning a higher-quality student model than with existing KD methods (Section 6), scales student's performance with teacher's size even under large capacity gaps (Section 6.3.2), and suppresses mode-averaging and mode-collapse issues theoretically and empirically (Section 4 and 6.3.3).

Our main contributions to this paper are as follows:

- We introduce TAID (Section 3), a new knowledge distillation method that reimagines the distillation process as a dynamic, adaptive knowledge transfer from student to teacher distributions. This approach addresses common challenges in distilling large language models.

- We provide a theoretical analysis of TAID (Section 4) with a regression model as a proxy to the language modeling objective, demonstrating its ability to prevent mode collapse in the distillation process. This theoretical guarantee sets TAID apart from traditional self-distillation methods, which can suffer from mode collapse.

- We conduct extensive experiments (Section 6) across various model sizes and architectures, demonstrating TAID's superiority in both instruction tuning and pre-training scenarios. Moreover, we experimentally reveal TAID's robustness to capacity gaps (Section 6.3.2), and its ability to balance between mode averaging and mode collapse, unlike existing KD methods (Section 6.3.3).

- We demonstrate TAID's practical impact by developing two state-of-the-art compact models (Section 7): `TAID-LLM-1.5B` achieves the best performance for language models under 2B parameters, while `TAID-VLM-2B` outperforms vision-language models up to 4B parameters, showcasing TAID's effectiveness across different domains.

## 2 PRELIMINARIES

**Problem setting for language model distillation.** A language model is defined as a probability distribution $p$ over token sequences $\mathbf{y} = (y_1, y_2, \ldots, y_S) \in \mathcal{Y}^S$, where $\mathcal{Y}$ is the vocabulary and $S$ is the sequence length. The distribution is obtained by applying the softmax function to logit values: $p(y_s \mid y^{<s}) = \text{softmax}(\text{logit}_p(y_s \mid y^{<s})) = \exp(\text{logit}_p(y_s|y^{<s}))/\sum_{y' \in \mathcal{Y}} \exp(\text{logit}_p(y'|y^{<s}))$. The model satisfies the autoregressive property: $p(\mathbf{y}) = \prod_{s=1}^{S} p(y_s \mid y^{<s})$ where $y^{<s} := (y_1, y_2, \ldots, y_{s-1})$, and $p(y_s \mid y^{<s}) = p(y_1)$ for $s = 1$. In KD for language models, we aim to transfer knowledge from a well-trained teacher model $p$ to a parametric student model $q_\theta$. The objective is to find parameters $\theta$ that minimize a distance measure $J$ between their distributions.

**Traditional knowledge distillation approaches.** Hinton et al. (2015) introduced KD using the Kullback–Leibler (KL) divergence, which is formulated for language models as: $J_{\text{KL}}(p, q_\theta) := \frac{1}{S} \sum_{s=1}^{S} \sum_{y_s \in \mathcal{Y}} p(y_s \mid y^{<s}) \log \frac{p(y_s|y^{<s})}{q_\theta(y_s|y^{<s})}$. However, KD based on the standard KL divergence often suffers from the *mode-averaging* problem, where a student model attempts to aggressively cover all modes of a teacher distribution despite being incapable, potentially resulting in a over-smoothed and less accurate distribution (Wen et al., 2023; Gu et al., 2024). To address this, Wen et al. (2023) proposed using the Reverse KL (RKL) divergence: $J_{\text{RKL}}(p, q_\theta) := J_{\text{KL}}(q_\theta, p)$. While this approach mitigates the mode-averaging problem, it can lead to *mode collapse*, where the student model focuses only on the dominant modes of the teacher distribution.

**Curse of capacity gap.** Mirzadeh et al. (2020), Cho & Hariharan (2019), and Zhang et al. (2023b) reported a curse of capacity gap, where an excessively large model can negatively impact the performance of the student model. This phenomenon poses a significant challenge in KD, particularly for language models. As state-of-the-art language models continue to grow in size and complexity, the capacity gap becomes increasingly critical in developing high-performing and compact student models. Addressing the capacity gap is crucial for effectively transferring knowledge from large-scale language models to more portable ones without sacrificing performance. Our experiments (Section 6.3.2) provide empirical evidence of the capacity gap and demonstrate how our proposed method addresses this challenge.

## 3 PROPOSED METHOD: TAID

We introduce Temporally Adaptive Interpolated Distillation (TAID), a novel knowledge distillation method for large language models. TAID uses a dynamic, time-dependent intermediate teacher to bridge the gap between student and teacher models (see Figure 1). This approach facilitates smoother knowledge transfer, addressing the capacity gap and balancing mode-averaging and mode-collapse issues. We show how TAID mitigates these issues in Sections 6.3.2 and 6.3.3, respectively.

### 3.1 TEMPORALLY INTERPOLATED DISTRIBUTION

The key idea behind TAID is to employ a time-dependent intermediate teacher to bridge the gap between student and teacher models. We formally define the intermediate distribution as follows:

**Definition 3.1** (TAID Interpolated Distribution). *For any input sequence $y^{<s} \in \mathcal{Y}^{s-1}$ and any output token $y_s \in \mathcal{Y}$, the TAID interpolated distribution $p_t$ is defined as:*

$$p_t(y_s|y^{<s}) := \text{softmax}\left((1-t) \cdot \text{logit}_{q'_\theta}(y_s|y^{<s}) + t \cdot \text{logit}_p(y_s|y^{<s})\right) \quad (1)$$

*where $t \in [0, 1]$ is a time-dependent interpolation parameter, $\text{logit}_{q'_\theta}$ represents a detached version of the student logits (i.e., treated as a constant without being backpropagated), and $\text{logit}_p$ represents the teacher logits.*

The interpolation is performed at the logit level to preserve relative confidence between predictions. The TAID objective function with the interpolation parameter $t$ is defined as the KL divergence between the intermediate distribution $p_t$ and the student distribution $q_\theta$:

**Definition 3.2** (TAID Objective). *The TAID objective function at time $t$ is defined as:*

$$J_{\text{TAID}}^{(t)}(p, q_\theta) := J_{\text{KL}}(p_t, q_\theta) = \frac{1}{S} \sum_{s=1}^{S} \sum_{y_s \in \mathcal{Y}} p_t(y_s|y^{<s}) \log \frac{p_t(y_s|y^{<s})}{q_\theta(y_s|y^{<s})}. \tag{2}$$

We gradually increase the interpolation parameter $t$ from 0 to 1 during training so that the intermediate distribution $p_t$ adaptively transitions from the student's initial distribution towards the teacher's distribution. Refer to Section 3.2 for the scheduling of the interpolation parameter. The detached $q'_\theta$ in $p_t$ ensures that we only optimize the student model $q_\theta$ in the denominator of the KL divergence, effectively treating the intermediate distribution as a target.

The intermediate distribution provides a crucial advantage in addressing the capacity gap and mode-averaging/collapse issues. By smoothly transitioning from the student's initial distribution to the teacher's distribution, TAID facilitates a gradual transfer of knowledge. This approach effectively mitigates issues associated with significant capacity gaps between teacher and student models. This can be understood as follows: When $t$ is small, the student model is encouraged to focus on its own modes, reinforcing its unique characteristics. In this phase, TAID behaves similarly to self-distillation (using the student model as the teacher), which amplifies generalization by sparsifying the model (Mobahi et al., 2020). Thus, the student model tends to capture dominant features of the student's distribution. As $t$ increases, the student gradually incorporates the teacher's knowledge, capturing more nuanced and rich signals from the teacher distribution. This balanced approach results in a student model that not only captures the essential knowledge from the teacher but also maintains its ability to generalize effectively. Despite TAID's relevance to self-distillation, the interpolation parameter is essential to avoid mode collapse, which self-distillation cannot escape. We will theoretically demonstrate it in Section 4.

## 3.2 ADAPTIVE INTERPOLATION PARAMETER UPDATE

While TAID demonstrates effectiveness even with a simple linear increase of the interpolation parameter $t$, we propose an adaptive update mechanism to achieve more efficient learning and improved accuracy. The key motivation is to dynamically adjust $t$ based on the student's learning progress. The adaptive update strategy is designed to aggressively increase $t$ in the early stages when the interpolated distribution $p_t$ is close to the student model $q_\theta$, as the model fitting is not challenging in this phase. As the student model approaches the teacher model, the increase in $t$ becomes more gradual, allowing for careful fitting to the more complex teacher distribution.

Our adaptive update strategy is based on the relative change in the objective function: $\delta_n := (J_{\text{TAID}}^{(t_{n-1})} - J_{\text{TAID}}^{(t_n)})/(J_{\text{TAID}}^{(t_{n-1})} + \epsilon)$, where $J_{\text{TAID}}^{(t_n)}$ is the value of the TAID objective function at interpolation parameter $t_n$, $t_n$ is the interpolation parameter at step $n$, and $\epsilon$ is a small constant to prevent division by zero. We update $t_n$ using a momentum-based approach to smooth out short-term fluctuations: $m_n = \beta m_{n-1} + (1 - \beta)\delta_n$, where $\beta$ is the momentum coefficient. The interpolation parameter is then updated as: $t_n \leftarrow \min(1.0, \max(t_{\text{linear}}, t_{n-1} + \alpha \cdot \text{sigmoid}(m_n) \cdot (1 - t_{n-1})))$, where $\alpha$ is the step size for $t$, and $t_{\text{linear}}$ is a linear increase schedule as a lower bound for $t$. To allow for flexible initialization, $t$ is set to a start value $t_{\text{start}}$, which is a hyperparameter. The complete TAID training procedure is summarized in Algorithm 1 in Appendix A.

This update mechanism allows for more aggressive increases in $t$ during the early stages of training when the student is learning rapidly (high $\delta_t$), and more gradual increases as the student model approaches the teacher's complexity (low $\delta_t$). The sigmoid function bounds the update, ensuring stable learning, while the max and min operations guarantee a monotonic increase within the predefined range. A detailed analysis of how different $\alpha$ values affect the behavior of $t$ and the learning dynamics is presented in Section 6.3.1.

## 4 THEORETICAL ANALYSIS

TAID distills from the intermediate distribution $p_t$, partially containing the student model $q_\theta$ as the mixture component. This may apparently cause the collapse because student's modes are amplified repeatedly during the fitting recursion. Such a collapse phenomenon has been theoretically observed

for self-distillation, where the teacher and student models are identical (Mobahi et al., 2020). We aim to demonstrate that TAID avoids mode collapse, unlike self-distillation.

We borrow the analysis framework of Mobahi et al. (2020) to study least-square regression as a proxy to language modeling. In each training step, the student model is updated by fitting to the interpolated label $(1 - t)y_t + ty_{\text{teacher}}$, where $y_t$ and $y_{\text{teacher}}$ are the labels of the current student and teacher models, respectively, and $t$ is the interpolation parameter (being linearly increased) at the current step. Here, we suppose the student model achieves $\epsilon$-*interpolation* of the training signals so that the regression loss is minimized near-perfectly in each time step.

**Theorem 4.1** (Non-collapse Nature (Informally)). *Suppose we run distillation for $T$ steps in total. If the teacher model has sufficiently large signals so that the label is at least as large as $\Omega(\sqrt{T\epsilon})$, then the student model does not collapse for any time $t$.*

Notably, self-distillation inevitably collapses for sufficiently large steps (Mobahi et al., 2020, Proposition 4), corroborating the benefit of the intermediate distribution and its adaptive update. The formal statement and more discussions can be found in Appendix B.

## 5    RELATED WORKS

**Improving objective functions.** To address the mode-averaging and mode-collapse issues that the traditional KL divergence-based methods (Section 2) face, various alternative objective functions have been applied to knowledge distillation. Wen et al. (2023) applied the Total Variation Distance, formulated at the sequence level similar to Kim & Rush (2016): $J_{\text{TVD}}(p, q_\theta) := \frac{1}{2}\sum_{\mathbf{y}}|p(\mathbf{y}) - q_\theta(\mathbf{y})|$. Agarwal et al. (2024) utilized the Generalized Jensen–Shannon (JS) Divergence: $J_{\text{GJSD}}(p, q_\theta) := \lambda J_{\text{KD}}(p, r) + (1 - \lambda)J_{\text{RKD}}(p, r)$, where $r(\mathbf{y}) = \lambda p(\mathbf{y}) + (1 - \lambda)q_\theta(\mathbf{y})$ and $\lambda \in [0, 1]$. Additionally, Ko et al. (2024) employed the Skew KL Divergence: $J_{\text{SKD}}(p, q_\theta) := J_{\text{KL}}(p, r)$. They also defined the Skew Reverse KL Divergence as $J_{\text{SRKD}}(p, q_\theta) := J_{\text{KL}}(q_\theta, r)$. These approaches aim to balance preserving teacher knowledge and allowing student generalization. However, they typically use a fixed teacher distribution throughout distillation, potentially hindering knowledge transfer when there is a significant capacity gap between teacher and student. In contrast, our TAID method introduces a time-dependent intermediate distribution, gradually transitioning from the student's initial distribution to the teacher's, mitigating the capacity gap issue and enabling more stable learning. While Skew KL divergence also adopts an intermediate distribution, its approach differs significantly from TAID. Skew KL divergence uses a fixed intermediate distribution and transfers the teacher's knowledge to it, whereas TAID employs a time-dependent intermediate distribution and transfers it to the student. This distinction, particularly the dynamic nature of TAID's intermediate distribution, makes TAID more suitable for adaptive updates of the student model as the interpolation parameter changes over time (see Appendix C for a detailed comparison).

**Utilizing student-generated outputs (SGOs).** Recent research in KD for language models has explored utilizing on-policy data sampled from teacher and student models during training (Gu et al., 2024; Zhang et al., 2024b). Within this approach, some studies have specifically focused on leveraging student-generated outputs (SGOs) (Agarwal et al., 2024; Ko et al., 2024). While these methods show promise in improving distillation performance and addressing the distribution mismatch between training and inference due to the autoregressive nature of LMs when training on a fixed dataset (Pomerleau, 1991; Ross & Bagnell, 2010), they are computationally expensive for large-scale models. TAID achieves superior performance without relying on on-policy data or SGOs, offering improved computational efficiency for large-scale datasets and models (see Section 6.1). Future work could explore combining TAID with on-policy approaches to potentially achieve even better performance.

**KD methods from image classification.** KD has been extensively studied in image classification tasks, with some logit-based methods being applicable to language model distillation. Notable examples include CTKD (Li et al., 2023b) and DKD (Zhao et al., 2022), which have shown remarkable performance using standard KL divergence. CTKD shares a similar curriculum learning approach with TAID, gradually increasing task difficulty. CTKD achieves this through a learnable temperature parameter that modifies both student and teacher distributions. In contrast, TAID modifies only the

Table 1: **Evaluating distillation methods for LLM instruction tuning.** The MT-Bench scores after training are listed, where higher scores indicate better conversational performance. For each of the three teacher-student pairs, different distillation algorithms, including the proposed TAID method, are compared. The highest score in each column is highlighted in **bold**.

| Method | Teacher Student | Phi-3-mini (3.8B) TinyLlama (1.1B) | Llama-2 (6.7B) TinyLlama (1.1B) | StableLM Zephyr (2.8B) Pythia (0.4B) |
|---|---|---|---|---|
| SFT | | 2.00 | 3.94 | 2.57 |
| KL (Hinton et al., 2015) | | 2.71 | 3.99 | 2.74 |
| RKL (Wen et al., 2023; Gu et al., 2024) | | 3.48 | 3.92 | 2.53 |
| TVD (Wen et al., 2023) | | 3.27 | 3.64 | 2.57 |
| Adaptive KL (Wu et al., 2024) | | 3.27 | 3.77 | 2.64 |
| GKD (Agarwal et al., 2024) | | 2.24 | 3.82 | 2.59 |
| DistiLLM (Ko et al., 2024) | | 3.23 | 3.97 | 2.97 |
| CTKD (Li et al., 2023b) | | 1.78 | 2.84 | 1.39 |
| DKD (Zhao et al., 2022) | | 2.70 | 4.14 | 2.90 |
| **(Ours) TAID w/o adaptive update** | | 3.44 | 4.18 | 2.88 |
| **(Ours) TAID** | | **4.05** | **4.27** | **3.05** |

teacher distribution through interpolation, potentially preserving more of the student's learned information. DKD decomposes KL divergence into target and non-target class components, allowing for better weight adjustment in tasks of varying difficulty. However, these image classification-based methods are not sufficiently effective in language modeling due to the unique characteristics of the language domain. We experimentally verified it in Section 6.3.4. TAID addresses these challenges through its adaptive interpolation, while remaining flexible enough to be combined with methods like DKD for simpler tasks.

## 6 EMPIRICAL ANALYSIS

We evaluate TAID across instruction tuning and pre-training scenarios, using various model sizes and architectures. Our experiments compare TAID against state-of-the-art methods, demonstrating its superior performance and efficiency, while providing insights into its behavior across different capacity gaps and its ability to balance mode-averaging and mode-collapse issues.

### 6.1 INSTRUCTION TUNING

**Experimental setup.** For the instruction-following task, we used the UltraChat 200k dataset (Ding et al., 2023) for training. Performance was assessed using MT-Bench (Zheng et al., 2023), a benchmark designed to evaluate model's instruction-following ability, with scoring conducted by GPT-4. For our experiments, we utilized three teacher-student pairs: `Phi-3-mini-4k-instruct` (Abdin et al., 2024) as teacher with `TinyLlama` (Zhang et al., 2024a) as student, `Llama-2-7b-chat` (Touvron et al., 2023) as teacher with `TinyLlama` as student, and `StableLM Zephyr 3B` (Team, 2023) as teacher with `Pythia-410M` (Biderman et al., 2023) as student. To evaluate the pure effectiveness of our distillation method, we focused solely on distillation using instruction data, unlike previous studies (Gu et al., 2024; Agarwal et al., 2024; Ko et al., 2024) that often perform supervised fine-tuning (SFT) before distillation or include additional cross-entropy loss on pre-training corpora. Furthermore, to simulate a more practical scenario, we used powerful teacher models trained on in-house data with open weights for distillation to smaller student models. We compared TAID against prior works, including KL divergence (Hinton et al., 2015), RKL (Wen et al., 2023), Total Variation Distance (TVD) (Wen et al., 2023), Adaptive KL (Wu et al., 2024), as well as methods utilizing SGOs such as Generalized KD (GKD) (Agarwal et al., 2024) and DistiLLM (Ko et al., 2024). Additionally, we included two methods originally proposed for image classification tasks: CTKD (Li et al., 2023b) and DKD (Zhao et al., 2022), to assess their effectiveness in language model distillation. We also included a supervised fine-tuning (SFT) baseline to demonstrate the benefits of knowledge distillation. To isolate the impact of our adaptive update mechanism, we evaluated TAID both with and without this feature, where TAID without adaptive update uses a linear increase of the interpolation parameter with respect to the

Table 2: **Evaluating distillation methods for LLM continued pre-training**. The Open LLM Leaderboard scores after training are listed, with higher scores indicating better performance. The average score across the 6 tasks (Average column) is commonly used as an indicator of overall language proficiency. The highest score in each column is highlighted in **bold**.

| Method | ARC | HellaSwag | MMLU | TrustfulQA | Winogrande | GSM8K | Average |
|---|---|---|---|---|---|---|---|
| SFT | 41.38 | 63.66 | 25.89 | 35.64 | 61.25 | 1.21 | 38.17 |
| KL (Hinton et al., 2015) | 44.97 | 65.43 | 25.11 | **37.95** | 63.22 | 2.80 | 39.91 |
| TVD (Wen et al., 2023) | 43.52 | 64.50 | 25.95 | 36.38 | 63.14 | 2.96 | 39.41 |
| Adaptive KL (Wu et al., 2024) | 43.77 | 63.09 | 26.04 | 36.42 | 63.22 | 2.12 | 39.11 |
| GJS (Agarwal et al., 2024) | 44.71 | **65.67** | 25.27 | 37.76 | 62.12 | 3.34 | 39.81 |
| Skew KL (Ko et al., 2024) | 44.62 | 65.25 | 25.79 | 37.45 | 62.51 | **3.41** | 39.84 |
| Skew RKL (Ko et al., 2024) | 44.11 | 64.80 | **26.07** | 36.76 | 62.83 | 3.03 | 39.60 |
| **(Ours) TAID** | **45.48** | 65.43 | 25.43 | 37.92 | **63.38** | 2.96 | **40.10** |

training steps. Detailed hyper-parameters and implementation specifics for TAID and all baseline methods are provided in Appendix D.1.

**Results.** Table 1 presents the MT-Bench scores for all methods across the three different teacher-student pairs. Our proposed TAID method consistently outperforms all baseline methods, including those proposed for image classification (CTKD and DKD) and methods utilizing SGOs such as GKD and DistiLLM. Notably, TAID achieves superior performance without relying on expensive SGO sampling strategies, resulting in significantly faster training times—approximately 2 times faster than DistiLLM and 10 times faster than GKD. This combination of superior performance and computational efficiency, achieved without SGOs, makes TAID particularly attractive for real-world applications where both model quality and training speed are crucial. An ablation study comparing TAID with and without adaptive updates shows improvements ranging from 2.2% to 17.7% across different teacher-student pairs, underlining the importance of our proposed adaptive mechanism.

## 6.2 PRE-TRAINING

**Experimental setup.** Due to the limited resources, we performed continued pre-training, initializing the student model with a pre-trained model and further refining it through additional pre-training using distillation. We used the first 10% of the SmolLM-Corpus (Ben Allal et al., 2024) dataset, amounting to approximately 20 billion tokens. We used `Phi-3-medium-4k-instruct` (Abdin et al., 2024) as the teacher model and `TinyLlama` as the student model. Similar to our instruction tuning experiments, we focused solely on distillation without additional supervised fine-tuning or pre-training losses. Due to the computational cost associated with sampling from the student model in large-scale pre-training and the absence of prompts as in instruction-following tasks, we adapted the baseline methods to use only their objective functions without SGOs. We compared TAID against these modified baselines, including KL divergence, TVD, Adaptive KL, GJS (used in GKD), and Skew KL/RKL (used in DistiLLM). To evaluate the pre-trained models, we followed the Open LLM Leaderboard (Beeching et al., 2023) methodology, which is commonly used to assess the underlying capabilities of models through few-shot evaluation. This methodology includes six diverse tasks, with evaluation settings and metrics adhering to the Open LLM Leaderboard standards. Detailed hyperparameters and implementation specifics are provided in Appendix D.2.

**Results.** Table 2 presents the results of our pre-training experiments. Following the standard practice in the LLM community, we reported the average scores across diverse tasks. TAID achieves the highest average score across all six tasks, outperforming all baseline methods. This superior average performance demonstrates TAID's effectiveness in transferring knowledge from the teacher to the student model across a diverse range of tasks. While TAID shows the best overall performance, it is worth noting that it achieves the highest scores on two individual tasks (ARC and Winogrande) and competitive performance on the others. The consistently strong performance across tasks, coupled with the highest average score, underscores TAID's robustness and effectiveness in knowledge distillation for large language models.

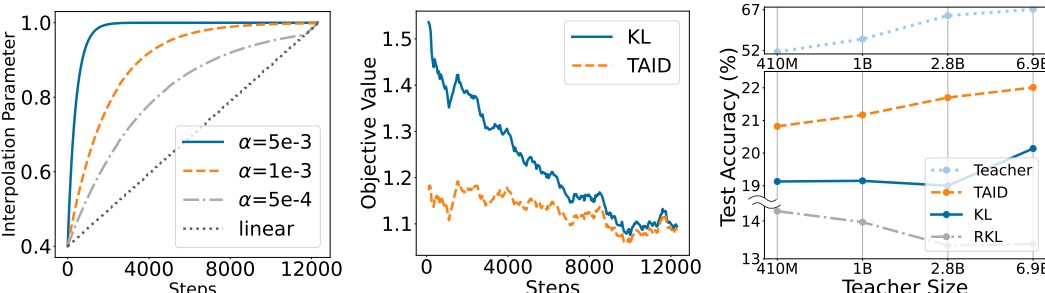

Figure 2: **Analysis of TAID's behavior and performance. (Left)** Interpolation parameter $t$ behavior: Higher $\alpha$ values lead to faster initial growth compared to linear increase, allowing for more aggressive knowledge transfer in early stages when the capacity gap is small. **(Middle)** Objective value comparison: TAID exhibits a more stable objective value with lower variance compared to standard KL divergence throughout training, indicating a consistent learning difficulty that aligns with the student's evolving capabilities. **(Right)** Performance across different teacher sizes: TAID shows monotonic improvement and outperforms other methods as teacher size increases, demonstrating its effectiveness in addressing the curse of capacity gap.

## 6.3 ANALYSIS

### 6.3.1 ANALYSIS OF INTERPOLATION PARAMETER AND TRAINING STABILITY

We analyzed TAID's interpolation parameter $t$ and learning dynamics to validate its design. Figure 2 (Left) shows how different learning rates $\alpha$ affect $t$'s behavior over time under the setting of Section 6.1, with $t_{\text{start}}$ set to 0.4. We can confirm that $t$ is smoothly increasing thanks to our adaptive update mechanism. Higher $\alpha$ values lead to faster initial growth of $t$, enabling more aggressive early knowledge transfer, which is particularly beneficial when the capacity gap between student and teacher models is small.

Figure 2 (Middle) compares the objective value of TAID (using the intermediate distribution) with the standard KL divergence between the teacher and student during training. TAID demonstrates a constant value with low variance throughout the training process, in contrast to the higher and more variable loss of standard KL. This stability in loss indicates that TAID's adaptive interpolation mechanism keeps the learning task at a consistent level of difficulty, aligning with the student's current capabilities. This controlled learning environment potentially leads to more efficient and stable knowledge transfer throughout the training process.

### 6.3.2 PERFORMANCE ACROSS VARIOUS CAPACITY GAPS

TAID's design, which gradually transfers knowledge from the teacher model, is expected to address the curse of capacity gap described in Section 2. To evaluate this, we conducted an experiment using a fixed-size student model (70m) trained with teachers of varying capacities (410M to 6.9B) from the Pythia Suite (Biderman et al., 2023). Models were trained on a random 1B token subset of the SmolLM-Corpus for 1 epoch, due to computational cost constraints. We chose the LAMBADA dataset (Paperno et al., 2016) for evaluation, as it tests a model's ability to predict the final word of a passage, directly assessing language modeling capability without relying on specific knowledge, making it suitable for comparing models with small-scale training.

Figure 2 (Right) shows that TAID consistently outperforms both KL and RKL divergence methods across all teacher model sizes. Notably, TAID exhibits a consistent upward trend in performance as the teacher model size increases while KL and RKL methods show inconsistent performance trends. This inconsistency in KL and RKL methods aligns with the curse of capacity gap, where larger teacher models do not always lead to better student performance, described Section 2. TAID's consistent improvement with larger teachers indicates its robustness in handling varying capacity gaps, making it particularly suitable for distilling knowledge from state-of-the-art large language models into more compact and deployable student models.

### 6.3.3 BALANCING MODE AVERAGING AND MODE COLLAPSE

To demonstrate TAID's effectiveness in balancing mode-averaging and mode-collapse issues, we analyzed the distributions of student models trained using KL divergence, RKL divergence, and TAID. We used the trained models of the `Phi-3-mini-4k-instruct` (teacher) and `TinyLlama` (student) pair in Section 6.1, with distributions calculated from the UltraChat 200k train set.

Table 3 presents a summary of our analysis, showing the probability mass distribution for the head and tail of the vocabulary as ranked by the teacher model. We observe that TAID consistently maintains probability masses between those of KL and RKL for both the head and tail of the distribution.

In the head, TAID captures dominant vocabulary in the teacher's distribution more than KL, effectively avoiding the mode-averaging issue. While RKL captures the dominant vocabulary more than TAID, it significantly fails to capture low-frequent vocabulary in the tail of the teacher distribution, which TAID captures reasonably, preventing the mode-collapse issue. These results indicate that TAID successfully navigates the trade-off between mode averaging and mode collapse, achieving a more balanced and faithful representation of the teacher's distribution across both common and rare tokens. This balanced approach contributes to TAID's superior performance in knowledge distillation tasks, as it more effectively captures the full spectrum of the teacher's knowledge while maintaining a focused distribution.

Table 3: **Probability mass distribution analysis.** Head: sum of probabilities for top-10 tokens. Tail: sum of probabilities for tokens in the 80–100th percentile.[1]

| Method | Head | Tail |
|--------|------|------|
| KL | 0.216 | $40.2 \times 10^{-7}$ |
| RKL | 0.227 | $8.1 \times 10^{-7}$ |
| TAID | 0.218 | $39.0 \times 10^{-7}$ |

### 6.3.4 COMPARISON WITH IMAGE CLASSIFICATION TASKS

Our experiments revealed that KD methods developed for image classification, such as CTKD (Li et al., 2023b) and DKD (Zhao et al., 2022), underperform in language model distillation. We hypothesize that this is due to fundamental differences in the distributions between language modeling tasks and image classification tasks. Figure 3 illustrates the entropy of the distribution and the probabilities of ground-truth classes (target-class probabilities) for two representative models: ResNet-56 (He et al., 2016) for image classification and GPT-2 (Radford et al., 2019) for language modeling.[2] Image classification typically involves predicting a one-hot distribution with high target-class probability and low entropy. In contrast, language modeling predicts a more diverse probability distribution, resulting in lower target-class probabilities and higher entropy. These characteristics lead to two key challenges in language model distillation. First, there is an increased susceptibility to mode collapse, as the model can easily be pulled toward non-target modes. Second, language modeling poses a significant challenge for smaller models with limited capacity: predicting extremely low-frequency classes. This difficulty is compounded by a power law distribution of word frequencies (Zipf's law), resulting in a large number of extremely low-frequency classes in the long tail of the distribution. To test this hypothesis and to assess TAID's flexibility, we evaluated TAID on multiple image classification tasks (results in Appendix D.3). While gains were modest on CIFAR-100, TAID consistently outperformed CTKD and DKD on the more complex ImageNet task. This aligns with our observation that ImageNet (entropy: 6.67, target-class probability: 0.00130) presents a more challenging distribution compared to CIFAR-100 (entropy: 0.485, target-class probability: 0.613). These findings highlight the need for distillation methods tailored to language modeling's unique challenges. TAID's strong performance

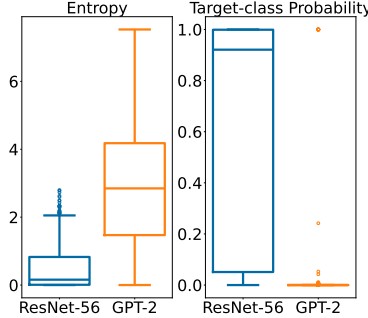

Figure 3: **Comparison between image classification and language modeling tasks.** Language modeling (GPT-2) exhibits significantly higher entropy and lower target-class probabilities compared to image classification (ResNet-56). These fundamental differences highlight the unique challenges in language model distillation.

---

[1]Typically, probabilities range from $10^{-1}$ to $10^{-2}$ for Head tokens and from $10^{-10}$ to $10^{-11}$ for Tail tokens.

[2]For this analysis, we used the CIFAR-100 (Krizhevsky, 2009) dataset for ResNet-56 and the OpenWeb-Text (Gokaslan & Cohen, 2019) dataset for GPT-2.

Table 4: **Performance of `TAID-LLM-1.5B`**, our new state-of-the-art LLM for models under 2B parameters. See Table 9 for task breakdown.

| Model | LightEval (↑) |
|---|---|
| `Qwen2-1.5B` (Yang et al., 2024) | 46.19 |
| `Phi-1.5B` (Li et al., 2023a) | 50.39 |
| `StableLM-2-1.6B` (Bellagente et al., 2024) | 51.24 |
| `SmolLM-1.7B` (Allal et al., 2024) | 51.31 |
| **`TAID-LLM-1.5B`** | **52.27** |

Table 5: **Performance of `TAID-VLM-2B`**, our new state-of-the-art VLM for models up to 4B parameters. See Table 10 for task breakdown.

| Model | Open-VLM-LB (↑) |
|---|---|
| `PaliGemma` (Beyer et al., 2024) | 46.56 |
| `MiniCPM-V-2` (Yao et al., 2024) | 47.93 |
| `Phi-3-Vision` (Abdin et al., 2024) | 53.60 |
| `InternVL2-2B` (Chen et al., 2024) | 53.96 |
| **`TAID-VLM-2B`** | **56.43** |

across domains, particularly in complex tasks, demonstrates its potential as a versatile approach to knowledge distillation. Future work could explore its application to other tasks involving long-tail distributions or complex probability predictions beyond language modeling.

# 7 APPLICATION TO STATE-OF-THE-ART MODEL DEVELOPMENT

Building upon our systematic evaluation of TAID, we further demonstrate its effectiveness in developing state-of-the-art models. We introduce two models: `TAID-LLM-1.5B` and `TAID-VLM-2B`, which have achieved state-of-the-art performance in their respective size categories for large language models (LLMs) and vision-language models (VLMs).

**TAID-LLM-1.5B.** We developed `TAID-LLM-1.5B`, a new 1.5B-parameter language model, using our TAID method. Following recent conventions in evaluating language models of this size (Allal et al., 2024), we evaluated it using LightEval [3], a comprehensive benchmark suite for small language models. Table 4 shows that `TAID-LLM-1.5B` achieves the highest score, setting a new state-of-the-art for models with fewer than 2 billion parameters. Detailed settings and results can be found in Appendix E.1.

**TAID-VLM-2B.** To showcase TAID's versatility, we developed `TAID-VLM-2B`, a new 2B-parameter vision-language model. We evaluated it following the Open VLM Leaderboard protocol (OpenCompass Contributors, 2023)[4]. As shown in Table 5, `TAID-VLM-2B` achieves the highest score among state-of-the-art vision-language models up to 4B parameters, even surpassing the performance of larger models like `Phi-3-Vision` (4.2B parameters). This success highlights TAID's capability in transferring multimodal knowledge across significant capacity gaps. Detailed settings and results can be found in Appendix E.2.

# 8 CONCLUSION

We introduced Temporally Adaptive Interpolated Distillation (TAID), a novel knowledge distillation approach that effectively addresses the challenges of compressing large language models. Our experiments demonstrated TAID's superior performance across various model sizes and architectures, consistently outperforming state-of-the-art methods. The development of `TAID-LLM-1.5B` and `TAID-VLM-2B`, achieving state-of-the-art performance in their categories, underscores TAID's practical impact. TAID's dynamic bridge mechanism effectively mitigates mode-averaging and mode-collapse problems, leading to more stable and efficient training. These advantages contribute to more accessible deployment of advanced language technologies in resource-constrained environments. Future research could extend TAID to other distance metrics, explore non-linear interpolations, adapt it for multi-teacher distillation (Wan et al., 2024), and investigate its application in other modalities and tasks beyond classification. In conclusion, TAID represents a significant advancement in knowledge distillation, offering both theoretical insights and practical benefits. As AI evolves, techniques like TAID will be crucial in making these advancements more accessible and deployable in real-world applications.

---

[3] https://huggingface.co/blog/smollm
[4] https://huggingface.co/spaces/opencompass/open_vlm_leaderboard

## AUTHOR CONTRIBUTIONS

Makoto Shing and Takuya Akiba initiated this project. Makoto Shing is the main contributor who conceptualized and proposed the TAID method, designed and conducted all experiments, performed theoretical analysis, implemented the main code, wrote the initial draft of the manuscript, and was responsible for data analysis and interpretation of results. Consistently led and executed all aspects of the project from inception to completion. Kou Misaki contributed to data processing for the TAID-LLM-1.5B model. Han Bao provided crucial feedback on theoretical interpretations and analysis. Sho Yokoi offered valuable insights and feedback, especially based on his expertise in Natural Language Processing. Takuya Akiba served as the primary advisor throughout the project, offering guidance, technical insight, advice, and supervision from inception to completion. All authors reviewed and edited the final manuscript.

## ACKNOWLEDGEMENTS

The authors would like to thank Masanori Suganuma and Tianyu Zhao for providing valuable discussions and feedback while drafting the text. This work is based on results obtained from a project, JPNP20017, subsidized by the New Energy and Industrial Technology Development Organization (NEDO). This work was supported by JSPS KAKENHI (Grant No. 22H05106), JST FOREST (Grant No. JPMJFR2331), and JST PRESTO (Grant No. JPMJPR24K6).

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

## A  TAID TRAINING ALGORITHM

Algorithm 1 provides a detailed description of the TAID training procedure, including the adaptive update mechanism for the interpolation parameter $t$. The TAID algorithm utilizes several key

---

**Algorithm 1** TAID training algorithm

---

1: **Input:** Learning rate $\eta$, learning rate of the interpolation parameter $\alpha$, momentum coefficient $\beta$, total iterations $N$, start value $t_{\text{start}}$, end value $t_{\text{end}}$
2: Initialize student model parameters $\theta$
3: Initialize $t_1 = t_{\text{start}}$, $m_0 = 0$, $J_{\text{TAID}}^{(t_0)} = \infty$
4: **for** each training iteration $n = 1$ to $N$ **do**
5:     Compute linear increase value: $t_{\text{linear}} = t_{\text{start}} + (t_{\text{end}} - t_{\text{start}}) \cdot n/N$
6:     Sample batch $\{(y_j^{<s}, y_j)\}_{j=1}^{B}$ from dataset $\mathcal{D}$
7:     Compute $p_{t_n}(y_s|y^{<s})$ using Eq. (1)
8:     Compute $J_{\text{TAID}}^{(t_n)}$ using Eq. (2)
9:     Update $\theta$: $\theta \leftarrow \theta - \eta \nabla_\theta J_{\text{TAID}}^{(t_n)}$
10:     $\delta_n = (J_{\text{TAID}}^{(t_{n-1})} - J_{\text{TAID}}^{(t_n)})/(J_{\text{TAID}}^{(t_{n-1})} + \epsilon)$
11:     $m_n = \beta m_{n-1} + (1 - \beta)\delta_n$
12:     $\Delta t = \alpha \cdot \text{sigmoid}(m_n) \cdot (1 - t_n)$
13:     $t_{n+1} \leftarrow \min(t_{\text{end}}, \max(t_{\text{linear}}, t_n + \Delta t))$
14: **end for**

---

hyperparameters that control the behavior of the interpolation parameter $t$ and the adaptive update mechanism. We discuss the effects of these parameters below:

- $\alpha$ (learning rate of $t$): This parameter controls the speed of the adaptive update for $t$. Figure 2 (Left) shows the behavior of $t$ for different values of $\alpha$, including a linear increase for comparison. As $\alpha$ increases, we observe that $t$ grows more rapidly in the early stages when the student model is close to the initial interpolation distribution. This allows for more efficient learning when the task is relatively easy for the student.

- $\beta$ (momentum coefficient): This parameter controls the smoothness of the adaptive update. A higher value of $\beta$ results in more stable updates by reducing the impact of short-term fluctuations in the objective function. In our experiments, we found that a $\beta$ value around 0.99 worked well across different scenarios.

- $t_{\text{start}}$ (initial value of $t$): This parameter determines the starting point of the interpolation. It is particularly useful for skipping the initial stages of learning when the task is very easy for the student. The choice of $t_{\text{start}}$ should be based on the intuitive gap between the initial student and teacher models. In our experiments, we found that values between 0.2 and 0.4 often yield good results, depending on the initial similarity between the student and teacher models.

- $t_{\text{end}}$ (maximum value of $t$): This parameter sets the upper limit for $t$, typically set to 1.0 to ensure that the final distribution matches the teacher model.

The algorithm uses a linear increase schedule ($t_{\text{linear}}$) as a lower bound for $t$, ensuring that $t$ increases at least linearly over the course of training. This approach maintains the adaptive nature of TAID while guaranteeing a minimum rate of progression towards the teacher distribution.

In our experiments, TAID demonstrated robust performance across various tasks with minimal hyperparameter tuning. We usually used $\beta = 0.99$ and $\alpha = 5\mathrm{e}{-4}$, with $t_{\text{start}}$ typically ranging between 0.2 and 0.4, depending on the initial student-teacher similarity. While these default values often yield good results, practitioners may achieve further improvements by fine-tuning these parameters for their specific tasks and model architectures, particularly in cases that differ significantly from our experimental settings.

## B  THEORETICAL ANALYSIS OF MODE COLLAPSE

In this section, we formally study the mode-collapse behavior of TAID.

### B.1  ANALYSIS MODEL

To study the collapse phenomenon, we leverage the analysis framework used by Mobahi et al. (2020). We study the regression problem in the interpolation regime:[5]

$$f^* := \arg\min_{f \in \mathcal{F}} R(f) \quad \text{s.t.} \quad \frac{1}{N} \sum_{i=1}^{N} (f(\mathbf{x}_i) - y_i)^2 \leq \epsilon, \tag{3}$$

where $\mathcal{D} := \{(\mathbf{x}_i, y_i)\}_{i=1}^{N}$ is a finite training set with $d$-dimensional covariates $\mathbf{x}_i \in \mathcal{X} \subseteq \mathbb{R}^d$ and one-dimensional outcome $y_i \in \mathbb{R}$, $\epsilon > 0$ is a desired loss tolerance parameter, $R(f)$ is a regularization functional, and $\mathcal{F} \subseteq \mathbb{R}^{\mathcal{X}}$ is a hypothesis space. Since we are interested in a large model regime, $\mathcal{F}$ is reasonably assumed to be encompassing all measurable functions. The mean-squared loss is used in (3) instead of the KL divergence, which is convenient to obtain analytical solutions later. The regularizer in the following form is considered:

$$R(f) = \int u(\mathbf{x}, \mathbf{x}') f(\mathbf{x}) f(\mathbf{x}') \mathrm{d}\mathbf{x} \mathrm{d}\mathbf{x}', \tag{4}$$

where $u$ is a symmetric kernel inducing $R(f) \geq 0$ with equality only when $f = 0$. The interpolation problem (3) may *collapse* depending on the teacher signals. Let us stack labels into a vector:

$$\mathbf{y} := [y_1 \ y_2 \ \dots y_N]^\top \in \mathbb{R}^N.$$

When $\|\mathbf{y}\|^2 \leq N\epsilon$ holds, the problem (3) has a trivial solution $f = 0$. Such a collapse may happen particularly in the self-distillation paradigm because the teacher signals are (partially) given by our hypothesis itself. Thus, it is crucial to investigate when and whether the non-collapse condition $\|\mathbf{y}\|^2 > N\epsilon$ is satisfied to ensure that our hypothesis learns meaningful signals.

**Variational problem.**  The Lagrangian variational problem of (3) is given as follows:

$$f_\lambda^* := \arg\min_{f \in \mathcal{F}} \frac{1}{N} \sum_{i=1}^{N} (f(\mathbf{x}_i) - y_i)^2 + \lambda \int u(\mathbf{x}, \mathbf{x}') f(\mathbf{x}) f(\mathbf{x}') \mathrm{d}\mathbf{x} \mathrm{d}\mathbf{x}',$$

$$\text{where} \quad \frac{1}{N} \sum_{i=1}^{N} (f_\lambda^*(\mathbf{x}_i) - y_i)^2 - \epsilon = 0, \tag{5}$$

and $\lambda^{-1} > 0$ is the Lagrange multiplier. The solution to the variational problem (5) can be analytically written down. Let $g$ be the `Green function` of the linear operator $[Lf](\mathbf{x}) := \int u(\mathbf{x}, \mathbf{x}') f(\mathbf{x}') \mathrm{d}\mathbf{x}'$ such that

$$\int u(\mathbf{x}, \mathbf{x}') g(\mathbf{x}', \mathbf{x}_0) \mathrm{d}\mathbf{x}' = \delta(\mathbf{x} - \mathbf{x}_0), \tag{6}$$

---

[5]The *interpolation* regime must be distinguished from the time *interpolation* used in the proposed TAID.

where $\delta(\mathbf{x})$ is the Dirac delta. Let $\mathbf{G} \in \mathbb{R}^{N \times N}$ and $\mathbf{g_x} \in \mathbb{R}^N$ be

$$\mathbf{G}_{i,j} := \frac{1}{N} g(\mathbf{x}_i, \mathbf{x}_j) \quad \text{and} \quad \mathbf{g}_{\mathbf{x},i} := \frac{1}{N} g(\mathbf{x}, \mathbf{x}_i) \quad \text{for all } i, j \in [N].$$

Then, the analytical solution to (5) is given as follows (Mobahi et al., 2020, Proposition 1):

$$f_\lambda^*(\mathbf{x}) = \mathbf{g_x}^\top (\lambda \mathbf{I} + \mathbf{G})^{-1} \mathbf{y}. \tag{7}$$

If we diagonalize $\mathbf{G}$ (which is positive definite) as $\mathbf{G} = \mathbf{V}^\top \mathbf{D} \mathbf{V}$, the prediction vector over the training inputs $\mathbf{x}_1, \ldots, \mathbf{x}_N$ is given as

$$\mathbf{f} := [f_\lambda^*(\mathbf{x}_1) \ \ldots \ f_\lambda^*(\mathbf{x}_N)]^\top = \mathbf{V}^\top \mathbf{D}(\lambda \mathbf{I} + \mathbf{D})^{-1} \mathbf{V} \mathbf{y}. \tag{8}$$

The solution (8) is essentially a nonlinear extension of the ridge estimator. Note that $\mathbf{V} \in \mathbb{R}^{N \times N}$ is an orthogonal matrix and $\mathbf{D} = \text{diag}(d_1, \ldots, d_N)$ has positive eigenvalues solely.

Importantly, (7) is the solution to the variational problem (5), which is parametrized by $\lambda$ satisfying $\frac{1}{N} \sum_i (f_\lambda^*(\mathbf{x}_i) - y_i)^2 - \epsilon = 0$. Solving this in $\lambda$ is hard because of its non-linearity, but Mobahi et al. (2020, Eq. (24)) evaluate its upper and lower bound:

$$\lambda = \frac{\alpha \sqrt{N\epsilon}}{\|\mathbf{y}\| - \sqrt{N\epsilon}} \quad \text{for some } \alpha \in [d_{\min}, d_{\max}], \tag{9}$$

where $d_{\max} := \max_i d_i$ and $d_{\min} := \min_i d_i$. Thus, the analytical solution (7) with this range of $\lambda$ is a solution to the original interpolation problem (3), too.

**Remark on connection to language modeling.** The interpolation formulation (3) is based on the standard (one-dimensional) regression problem, which obviously deviates from the language modeling problem introduced in (2). Nonetheless, we believe that this formulation is not only beneficial for our transparent understanding owing to its simplicity but also has a connection to multi-categorical distributions. In distributional modeling, a student model $q_\theta$ outputs a probability distribution over $\mathcal{Y}$, and falls into `mode collapse` when $q_\theta$ has only few numbers of non-zero probabilities, that is, $\{c \in \mathcal{Y} \mid q_\theta(y = c) > 0\} \ll |\mathcal{Y}|$. To deal with the multi-categorical outputs, we can extend the one-dimensional problem (3) as follows:

$$\forall c \in \mathcal{Y}, \quad f_c^* := \arg\min_{f_c \in \mathcal{F}} R(f_c) \quad \text{s.t.} \quad \frac{1}{N} \sum_{i=1}^N (f_c(\mathbf{x}_i) - y_{i,c})^2 \le \epsilon,$$

where teacher signal $y_{i,c}$ is given in the one-hot format such that $\sum_{c \in \mathcal{Y}} y_{i,c} = 1$ and $y_{i,c} \in \{0, 1\}$ for all $c \in \mathcal{Y}$. We can follow the subsequent analysis straightforwardly. In this multi-categorical problem, a model $(f_c)_{c \in \mathcal{Y}}$ is regarded as falling into mode collapse if $f_c = 0$ for many $c \in \mathcal{Y}$. This is measured by the teacher signal condition $\|\mathbf{y}_c\|^2 \le N\epsilon$ for each $c$, where $\mathbf{y}_c \in \{0, 1\}^N$ is the stacked labels for class $c$. Thus, studying (3) is directly relevant to mode collapse in language modeling.

### B.2 FORMAL THEORETICAL STATEMENT

To study TAID in a fashion of the interpolation problem (3), we consider the following learning procedure listed in Algorithm 2. Here, the input signals $\mathbf{y}_0$ are deemed as the well-trained teacher—we can deem $\mathbf{y}_1$ as the well-trained teacher, but the resulting distillation dynamics would not change much.

**Theorem B.1.** *Let $\kappa := d_{\max}/d_{\min} (\ge 1)$ be the condition number of $\mathbf{G}$. The prediction vector $\mathbf{y}_{t+1}$ does not collapse, namely $\mathbf{y}_{t+1} = \mathbf{0}$ cannot be a solution to the interpolation problem (3), if for some $\gamma \in [0, 1]$, either of the following holds:*

$$t < \min\left\{ \frac{1}{\gamma + \kappa}(r_0 - \gamma) + o(1), \frac{\gamma}{r_0} T \right\} \quad \text{or} \quad \frac{1}{r_0} T < t, \tag{10}$$

*where $r_0 := \|\mathbf{y}_0\|/\sqrt{N\epsilon} > 1$ and $o(1)$ is an asymptotic term in the large $r_0$ limit.*

---

**Algorithm 2** TAID learning procedure for least-square regression

---

**Input:** $T$ number of iterations, $\mathbf{y}_0 \in \mathbb{R}^N$ input signals
1: $t \leftarrow 0$
2: **while** $t < T$ **do**
3: $\quad \tilde{\mathbf{y}}_t \leftarrow (1 - \frac{t}{T})\mathbf{y}_t + \frac{t}{T}\mathbf{y}_0$ $\qquad\qquad\qquad\qquad$ ▷ Compose intermediate teacher
4: $\quad \lambda_t \leftarrow \alpha_t\sqrt{N\epsilon}/(\|\tilde{\mathbf{y}}_t\| - \sqrt{N\epsilon})$ $\qquad\qquad$ ▷ Choose an appropriate $\lambda_t$ by (9)
5: $\quad \mathbf{y}_{t+1} \leftarrow \mathbf{V}^\top\mathbf{D}(\lambda_t\mathbf{I} + \mathbf{D})^{-1}\mathbf{V}\tilde{\mathbf{y}}_t$ $\quad$ ▷ Solve the variational problem with teacher $\tilde{\mathbf{y}}_t$ and $\lambda_t$
6: $\quad t \leftarrow t + 1$
7: **end while**

---

To make the asymptotics in $r_0$ work well, we need to ensure sufficiently strong initial signals $\|\mathbf{y}_0\|$ and/or near-interpolation (small $\epsilon$). The first bound in (10) is non-vacuous when $T = \Omega(r_0)$. Though it is a rather strong requirement, the asymptotic term becomes negligible numerically with a moderate magnitude of $r_0$ (like 5 to 10).

To see how TAID benefits from the intermediate teacher, compare the non-collapse condition (10) with that of self-distillation (Mobahi et al., 2020, Proposition 4):

$$t \leq \frac{r_0 - 1}{\kappa}. \tag{11}$$

We have two observations. First, TAID is beneficial in the latter phase of recursion (namely, step $t$ closer to $T$), where self-distillation can never escape from collapse eventually. This is an intuitive feature of TAID because the intermediate teacher partly consists strong signals $\mathbf{y}_0$ that does not depend on learned student predictors. Second, TAID is worse in the early phase of recursion (namely, step $t$ closer to 1) than self-distillation by a constant factor. Specifically, TAID and self-distillation have critical steps of collapse $t = O(r_0/(\gamma + \kappa))$ and $t = O(r_0/\kappa)$, respectively. To ensure that TAID learns meaningful features in the early phase, $\gamma$ should be reasonably bounded away from 0, leading to a worse critical point than self-distillation. This is a price that TAID has to pay for the stabilization in the latter phase.

By setting $\gamma = 1$ in (10), we get a more interpretable corollary, which is the formal version of Theorem 4.1.

**Corollary B.1.1.** *If initialization* $\|\mathbf{y}_0\|$ *satisfies*

$$\|\mathbf{y}_0\| = \Omega\left(\frac{1 + \sqrt{1 + 4T(1 + \kappa)}}{2}\sqrt{N\epsilon}\right),$$

*the prediction vector* $\mathbf{y}_{t+1}$ *does not collapse for any* $t$.

### B.3 PROOF

*Proof of Theorem B.1.* Subsequently, we use the change-of-variable $\mathbf{z}_t := \mathbf{V}\mathbf{y}_t$, where the norm is preserved $\|\mathbf{z}_t\| = \|\mathbf{y}_t\|$. We also write $\tilde{\mathbf{z}}_t := \mathbf{V}\tilde{\mathbf{y}}_t$ and $r_t := \|\tilde{\mathbf{z}}_t\|/\sqrt{N\epsilon}$ for convenience. At each time $t$, the non-collapse criterion is given by $\|\tilde{\mathbf{z}}_t\|^2 > N\epsilon(\iff r_t > 1)$: if it holds, the next update in Line 5 would not collapse. Let $\mathbf{A}_t := \mathbf{D}(\lambda_t\mathbf{I} + \mathbf{D})^{-1}$. We first show the second case, namely, the

prediction avoids collapse when $\frac{1}{r_0}T < t$. Then, $\tilde{\mathbf{z}}_t$ is recursively expanded.

$$
\begin{aligned}
\tilde{\mathbf{z}}_t &= \left(1 - \frac{t}{T}\right)\mathbf{z}_t + \frac{t}{T}\mathbf{z}_0 \\
&= \left(1 - \frac{t}{T}\right)\mathbf{A}_{t-1}\tilde{\mathbf{z}}_{t-1} + \frac{t}{T}\mathbf{z}_0 \\
&= \left(1 - \frac{t}{T}\right)\mathbf{A}_{t-1}\left[\left(1 - \frac{t-1}{T}\right)\mathbf{z}_{t-1} + \frac{t-1}{T}\mathbf{z}_0\right] + \frac{t}{T}\mathbf{z}_0 \\
&= \left(1 - \frac{t}{T}\right)\left(1 - \frac{t-1}{T}\right)\mathbf{A}_{t-1}\mathbf{z}_{t-1} + \left[\left(1 - \frac{t}{T}\right)\frac{t-1}{T}\mathbf{A}_{t-1} + \frac{t}{T}\mathbf{I}\right]\mathbf{z}_0 \\
&= \dots \\
&= \left[\prod_{\tau=0}^{t}\left(1 - \frac{t-\tau}{T}\right)\right]\cdot\left[\prod_{\tau=0}^{t-1}\mathbf{A}_\tau\right]\mathbf{z}_0 + \sum_{\tau=1}^{t-1}\left[\prod_{s=0}^{\tau-1}\left(1 - \frac{t-s}{T}\right)\right]\frac{t-\tau}{T}\left[\prod_{s=1}^{\tau}\mathbf{A}_{t-s}\right]\mathbf{z}_0 + \frac{t}{T}\mathbf{z}_0 \\
&= \left\{\frac{T!}{T^{t+1}\cdot(T-t-1)!}\left[\prod_{\tau=0}^{t-1}\mathbf{A}_\tau\right] + \sum_{\tau=1}^{t-1}\frac{(t-\tau)\cdot(T-t+\tau-1)!}{T^{\tau+1}\cdot(T-t-1)!}\left[\prod_{s=1}^{\tau}\mathbf{A}_{t-s}\right] + \frac{t}{T}\mathbf{I}\right\}\mathbf{z}_0 \\
&=: \overline{\mathbf{A}}_t\mathbf{z}_0.
\end{aligned}
\tag{12}
$$

To evaluate $\overline{\mathbf{A}}_t$, we first look at $\mathbf{A}_\tau$ for $\tau \in [0, t-1]$. Since $\mathbf{A}_\tau$ is a diagonal matrix, its $k$-th element of $\mathbf{A}_\tau$ can be expressed as follows:

$$
(\mathbf{A}_\tau)_k = \frac{d_k}{\lambda_\tau + d_k} = \left(\frac{\alpha_\tau/d_k}{\|\tilde{\mathbf{z}}_\tau\|/\sqrt{N}\epsilon - 1} + 1\right)^{-1}
\begin{cases}
\leq \left(\frac{1/\kappa}{\|\tilde{\mathbf{z}}_\tau\|/\sqrt{N}\epsilon - 1} + 1\right)^{-1} \leq 1 \\
\geq \left(\frac{\kappa}{\|\tilde{\mathbf{z}}_\tau\|/\sqrt{N}\epsilon - 1} + 1\right)^{-1} \geq 0
\end{cases},
\tag{13}
$$

where $\alpha_\tau$ is given in (9). The last inequalities can be formally shown by induction in $\tau \in [0, t-1]$. Thus, the minimum singular value of $\overline{\mathbf{A}}_t$ is evaluated as follows:

$$
\begin{aligned}
&\sigma_{\min}(\overline{\mathbf{A}}_t) \\
&= \sigma_{\min}\left(\frac{T!}{T^{t+1}\cdot(T-t-1)!}\left[\prod_{\tau=0}^{t-1}\mathbf{A}_\tau\right] + \sum_{\tau=1}^{t-1}\frac{(t-\tau)\cdot(T-t+\tau-1)!}{T^{\tau+1}\cdot(T-t-1)!}\left[\prod_{s=1}^{\tau}\mathbf{A}_{t-s}\right] + \frac{t}{T}\mathbf{I}\right) \\
&= \sigma_{\min}\left(\frac{T!}{T^{t+1}\cdot(T-t-1)!}\left[\prod_{\tau=0}^{t-1}\mathbf{A}_\tau\right]\right) + \sigma_{\min}\left(\sum_{\tau=1}^{t-1}\frac{(t-\tau)\cdot(T-t+\tau-1)!}{T^{\tau+1}\cdot(T-t-1)!}\left[\prod_{s=1}^{\tau}\mathbf{A}_{t-s}\right]\right) \\
&\quad + \sigma_{\min}\left(\frac{t}{T}\mathbf{I}\right) \\
&\geq \sigma_{\min}\left(\frac{t}{T}\mathbf{I}\right) \\
&= \frac{t}{T},
\end{aligned}
$$

where the second identity holds because all matrices evaluated are diagonal. This implies

$$
\|\tilde{\mathbf{z}}_t\| \geq \sigma_{\min}(\overline{\mathbf{A}}_t)\|\mathbf{z}_0\| \geq \frac{t}{T}\|\mathbf{z}_0\| = \frac{t}{T}\|\tilde{\mathbf{z}}_0\|.
$$

The last equality uses $\mathbf{z}_0 = \tilde{\mathbf{z}}_0$. Thus, the non-collapse criterion $\|\tilde{\mathbf{z}}_t\| > \sqrt{N}\epsilon$ holds as long as $t > (\sqrt{N}\epsilon/\|\tilde{\mathbf{z}}_0\|)T = (\sqrt{N}\epsilon/\|\mathbf{y}_0\|)T$.

Next, supposing $t$ is small enough such that $t \leq \frac{\gamma}{r_0}T$ with $\gamma \in (0,1)$, we show that the prediction avoids collapse when $t < (\frac{1}{2} + o(1))(r_0 - \gamma)$. To see the non-collapse criterion $r_t > 1$, we first

derive a lower bound of $r_t$:

$$r_t \overset{(12)}{=} \left\| \left(1 - \frac{t}{T}\right) \mathbf{A}_{t-1} \frac{\tilde{\mathbf{z}}_{t-1}}{\sqrt{N\epsilon}} + \frac{t}{T} \frac{\tilde{\mathbf{z}}_0}{\sqrt{N\epsilon}} \right\|$$

$$\overset{(a)}{\geq} \left(1 - \frac{t}{T}\right) \left\| \mathbf{A}_{t-1} \frac{\tilde{\mathbf{z}}_{t-1}}{\sqrt{N\epsilon}} \right\| - \frac{t}{T} \left\| \frac{\tilde{\mathbf{z}}_0}{\sqrt{N\epsilon}} \right\|$$

$$\geq \left(1 - \frac{t}{T}\right) \sigma_{\min}(\mathbf{A}_{t-1}) r_{t-1} - \frac{t}{T} r_0$$

$$\geq \left(1 - \frac{\gamma}{r_0}\right) \sigma_{\min}(\mathbf{A}_{t-1}) r_{t-1} - \gamma$$

$$\overset{(13)}{\geq} \left(1 - \frac{\gamma}{r_0}\right) \frac{r_{t-1}}{\frac{\kappa}{r_{t-1}-1} + 1} - \gamma$$

$$\overset{(b)}{\geq} \left(1 - \frac{\gamma}{r_0}\right) (\beta_0 r_{t-1} - \beta_1) - \gamma,$$

where (a) is due to the "reverse" triangle inequality and (b) is due to Mobahi et al. (2020, Eq. (137)) (which is essentially a linear lower bound of a convex function in $r_0$) with

$$\beta_0 := \frac{(r_0 - 1)^2 + \kappa(2r_0 - 1)}{(r_0 - 1 + \kappa)^2} \quad \text{and} \quad \beta_1 := \frac{r_0^2 \kappa}{(r_0 - 1 + \kappa)^2}.$$

By recursively lower bounding $r_t$, we obtain the following bound:

$$r_t \geq \left[\left(1 - \frac{\gamma}{r_0}\right) \beta_0\right]^t r_0 - \frac{\left(1 - \frac{\gamma}{r_0}\right) \beta_1 \left[\left(1 - \frac{\gamma}{r_0}\right)^t \beta_0^t - 1\right]}{\left(1 - \frac{\gamma}{r_0}\right) \beta_0 - 1} - \gamma =: \bar{\beta}_0^t r_0 - \bar{\beta}_1 \frac{\bar{\beta}_0^t - 1}{\bar{\beta}_0 - 1} - \gamma =: \underline{r}_t,$$

where $\bar{\beta}_0 := \left(1 - \frac{\gamma}{r_0}\right) \beta_0$ and $\bar{\beta}_1 := \left(1 - \frac{\gamma}{r_0}\right) \beta_1$. To derive the non-collapse condition, we solve $\underline{r}_t = 1$ to derive the critical $t$, which is equivalent to

$$t = \frac{\log\left(\frac{(1+\gamma)(1-\bar{\beta}_0) + \bar{\beta}_1}{\bar{\beta}_1 + r_0(1-\bar{\beta}_0)}\right)}{\log \bar{\beta}_0}.$$

By simple algebra,

$$t = \frac{\log\left(\frac{\gamma[r_0^2 + (\kappa-2)r_0 - (\kappa-1)] + (\kappa r_0^2 + \kappa(\kappa-1)r_0)}{\gamma^2[r_0 + 2(\kappa-1) - \frac{\kappa-1}{r_0}] + \gamma(\kappa-1)(\kappa+2-r_0 - \frac{1}{r_0}) + \kappa(\kappa-1+r_0^2)}\right)}{\log\left(\frac{1}{1-\frac{\gamma}{r_0}}\right) + \log\left(\frac{1}{1 - \frac{\kappa(\kappa-1)}{(r_0-1+\kappa)^2}}\right)}$$

$$\geq \frac{1 - \frac{\gamma^2[r_0 + 2(\kappa-1) - \frac{\kappa-1}{r_0}] + \gamma(\kappa-1)(\kappa+2-r_0 - \frac{1}{r_0}) + \kappa(\kappa-1+r_0^2)}{\gamma[r_0^2 + (\kappa-2)r_0 - (\kappa-1)] + (\kappa r_0^2 + \kappa(\kappa-1)r_0)}}{\left[\frac{1}{1-\frac{\gamma}{r_0}} - 1\right] + \left[\frac{1}{1 - \frac{\kappa(\kappa-1)}{(r_0-1+\kappa)^2}} - 1\right]}$$

$$= \frac{\frac{\kappa(\kappa-1)(r_0-1) + \gamma(r_0^2 + (2\kappa-3)r_0 - (\kappa-1)(\kappa+3) + \frac{\kappa-1}{r_0}) - \gamma^2[r_0 + 2(\kappa-1) - \frac{\kappa-1}{r_0}]}{\gamma[r_0^2 + (\kappa-2)r_0 - (\kappa-1)] + [\kappa r_0^2 + \kappa(\kappa-1)r_0]}}{\frac{1}{\frac{r_0}{\gamma} - 1} + \frac{1}{\frac{(r_0-1+\kappa)^2}{\kappa(\kappa-1)} - 1}}$$

$$= \frac{\frac{\gamma r_0^2 + [2\kappa-3-\gamma+\kappa(\kappa-1)]r_0 - (\kappa-1)[\kappa+\gamma(\kappa+3+2\gamma)] + \frac{\gamma(\kappa-1)(1+\gamma)}{r_0}}{(\gamma+\kappa)r_0^2 + [\gamma(\kappa-2)+(\kappa-1)]\kappa r_0 - \gamma(\kappa-1)}}{\frac{\gamma r_0^2 + (2\gamma+\kappa)(\kappa-1)r_0 - (\kappa+1)(\kappa-1)\gamma}{(r_0-\gamma)[r_0^2 + 2(\kappa-1)r_0 - (\kappa-1)]}}$$

where the inequality is due to $1 - \frac{1}{x} \leq \log x \leq x - 1$. The last lower bound can be asymptotically (in large $r_0$) expressed as follows:

$$t \geq \frac{\frac{\gamma + o(1)}{\gamma + \kappa + o(1)}}{\frac{\gamma + o(1)}{(r_0-\gamma)(1+o(1))}} = \frac{1}{\gamma+\kappa}(r_0 - \gamma) + o(1).$$

Table 6: **Performance comparison between TAID and Skew KL across different teacher sizes.** TAID shows consistent improvement with larger teachers, while Skew KL's performance degrades.

| Method | 410M | 1B | 2.8B | 6.9B |
|--------|------|------|------|------|
| TAID | 20.82 | 21.17 | 21.70 | 22.01 |
| SKL | 18.65 | 18.50 | 18.28 | 18.20 |

Thus, the non-collapse condition in the second case is $t < \frac{1}{\gamma+\kappa}(r_0 - \gamma) + o(1)$. $\qquad\square$

*Proof of Corollary B.1.1.* By the non-collapse criterion (10) with $\gamma = 1$,

$$\frac{1}{1+\kappa}(r_0 - 1) + o(1) \geq \frac{1}{r_0}T$$

suffices for $\mathbf{y}_t$ not being collapsed for any $t$. By solving this quadratic inequality, we can verify the statement. $\qquad\square$

## C    DETAILED COMPARISON WITH SKEW KL

We provide a detailed comparison between TAID and Skew KL to highlight their fundamental differences, focusing on two key aspects: the direction of knowledge flow and the nature of interpolation design.

The first key difference lies in the direction of knowledge flow, which can be understood through their objective functions. The TAID objective is formulated as $J_{\text{TAID}}(p, q_\theta) = J_{\text{KL}}(p_t, q_\theta)$, while the Skew KL objective takes the form $J_{\text{SKD}}(p, q_\theta) = J_{\text{KL}}(p, r)$, where $r(\mathbf{y}) = \lambda p(\mathbf{y}) + (1 - \lambda)q_\theta(\mathbf{y})$ and $\lambda \in [0, 1]$. In TAID, the interpolated distribution $p_t$ teaches the student model $q_\theta$, creating a direct path for knowledge transfer from the interpolated distribution to the student. Conversely, in Skew KL, the teacher $p$ teaches the interpolated distribution $r$, establishing an indirect path where the student's knowledge is mixed into the target distribution.

The second fundamental difference is in the design of the interpolation mechanism. TAID employs a time-dependent parameter $t$ that gradually changes during training, enabling adaptive knowledge transfer that evolves with the student's learning progress. In contrast, Skew KL uses a fixed interpolation parameter $\lambda$ throughout the training process, maintaining a constant mixing ratio between teacher and student distributions.

Our empirical study validates the benefits of these design choices, particularly in handling the capacity gap between teacher and student models. Table 6 shows the performance comparison across different teacher sizes, demonstrating that TAID achieves consistent improvement as teacher size increases from 410M to 6.9B parameters, while Skew KL's performance degrades with larger teachers.

## D    EXPERIMENTAL DETAILS

### D.1    INSTRUCTION TUNING EXPERIMENTS

For our instruction tuning experiments, we utilized the UltraChat 200k dataset. We preprocessed the dataset by removing samples exceeding a maximum length of $2048$ tokens, resulting in approximately 150k training samples and 2k validation samples.

All models were trained for 5 epochs using a batch size of $64$. We employed the AdamW optimizer with a learning rate of $1\text{e}{-}4$ and a cosine learning rate scheduler. To select the best checkpoint for evaluation, we calculated the ROUGE-L score on the validation set after each epoch and chose the checkpoint with the highest score.

For our proposed TAID method, we used a momentum coefficient ($\beta$) of $0.99$ across all experiments. The learning rate of $t$ ($\alpha$) was set to $5\text{e}{-}4$. The initial value of $t$ ($t_{\text{start}}$) was set to $0.4$ for the

Table 7: **Top-1 accuracies (%) on the CIFAR-100 dataset.** Results for different teacher-student pairs are shown.

| Method | Teacher
Student | ResNet56
ResNet20 | ResNet110
ResNet32 | ResNet32×4
ResNet8×4 | WRN-40-2
WRN-16-2 | WRN-40-2
WRN-40-1 | VGG13
VGG8 |
|---|---|---|---|---|---|---|---|
| KL (Hinton et al., 2015) | | 70.66 | 73.08 | 73.33 | 74.92 | 73.54 | 72.93 |
| CTKD (Li et al., 2023b) | | 71.19 | 73.52 | 73.39 | 75.45 | 73.93 | 73.52 |
| DKD (Zhao et al., 2022) | | 71.97 | 74.11 | 76.32 | 76.24 | 74.81 | 74.68 |
| MLKD (Jin et al., 2023) | | 72.19 | **74.11** | **77.08** | **76.63** | **75.35** | **75.18** |
| **(Ours) TAID** | | **72.25** | 73.51 | 74.85 | 75.81 | 74.51 | 74.38 |

`Phi-3-mini-4k-instruct` pair and $0.2$ for the other two pairs. The final value of $t$ ($t_{\text{end}}$) was set to $1.0$ for all experiments.

Regarding baseline methods, we implemented GKD using Generalized Jensen-Shannon Divergence (GJSD) with $\lambda = 0.1$ as the objective function and a student data fraction of $0.5$. For DistiLLM, we used Skew KL divergence with $\lambda = 0.1$ and an initial student data fraction of $0.0$. We selected the better performing skew divergence between Skew Forward KL and Skew Reverse KL based on the best ROUGE-L score. Following the original DistiLLM paper, we calculated the validation loss twice per epoch, totaling 10 times, to leverage the Adaptive SGO scheduler. For Adaptive KL, our implementation was used since no official implementation was available. For CTKD and DKD, we followed their settings used in the training on ImageNet (Deng et al., 2009).

In terms of computational efficiency, we observed significant differences in training times among the different methods. TAID completed its training in approximately 0.7 hours per epoch on our hardware setup using 8 NVIDIA H100 GPUs. In comparison, DistiLLM required about 2 hours per epoch, while GKD took approximately 9.8 hours per epoch under the same conditions. These differences in training time are primarily attributed to the computational complexity of methods utilizing SGOs. TAID's ability to achieve competitive performance without relying on SGOs contributes to its faster training times.

### D.2 PRE-TRAINING EXPERIMENTS

For our pre-training experiments, we used the first 10% of the SmolLM-Corpus (Ben Allal et al., 2024) dataset, which amounted to approximately 20 billion tokens.

The pre-training was conducted for 1 epoch using a distributed setup with 80 NVIDIA H100 GPUs, each processing a batch size of 8, resulting in an effective batch size of 640. We used the AdamW optimizer with a learning rate of $1e-4$ and a cosine learning rate scheduler.

The TAID-specific parameters for the pre-training experiments were kept consistent with those used in the `Phi-3- mini-4k-instruct` pair in the instruction tuning experiments. Also, the baseline methods in the pre-training experiments were implemented similarly to the instruction tuning experiments, with adjustments made to exclude SGOs due to the computational constraints of large-scale pre-training. Specifically, for methods like DistiLLM, we only used the core divergence components without the SGO-based additions.

### D.3 IMAGE CLASSIFICATION RESULTS

To explore TAID's applicability beyond language models, we conducted experiments on image classification tasks using the CIFAR-100 and ImageNet datasets.

### D.4 CIFAR-100 RESULTS

We evaluated TAID on the CIFAR-100 dataset, which consists of 100 classes. Table 7 presents the top-1 accuracies achieved by TAID and other knowledge distillation methods on various teacher-student model pairs.

Table 8: **Top-1 accuracies (%) on the ImageNet validation set.** Results for different teacher-student pairs are shown.

| Method | Teacher
Student | `ResNet34`
`ResNet18` | `ResNet50`
`MN-V1` |
|---|---|---|---|
| KD (Hinton et al., 2015) | | 71.03 | 70.50 |
| CTKD (Li et al., 2023b) | | 71.38 | 71.16 |
| DKD (Zhao et al., 2022) | | 71.70 | 72.05 |
| MLKD (Jin et al., 2023) | | 71.90 | **73.01** |
| **(Ours) TAID** | | **72.10** | 72.71 |

As shown in Table 7, TAID performs competitively on CIFAR-100, consistently outperforming KL divergence across all model pairs. However, the gains are modest compared to state-of-the-art methods specifically designed for image classification, such as MLKD.

Interestingly, based on the analysis of DKD, we can interpret that for simpler tasks like CIFAR-100, where the teacher's target class probabilities are close to 1, the weight of the NCKD component in DKD becomes small. This suggests that combining TAID with DKD could potentially lead to further performance improvements, leveraging the strengths of both approaches in handling different aspects of the distillation process.

### D.5 IMAGENET RESULTS

To assess TAID's performance on a larger-scale image classification task, we conducted experiments on the ImageNet dataset, which contains 1000 classes. Table 8 presents the top-1 accuracies achieved by TAID and other methods on ImageNet.

On ImageNet, TAID shows more pronounced improvements, consistently outperforming CTKD and DKD across both teacher-student pairs. For the ResNet34-ResNet18 pair, TAID achieves the highest accuracy among all methods. For the ResNet50-MobileNet-V1 pair, TAID performs competitively, outperforming CTKD and DKD, and achieving results close to MLKD.

These results on ImageNet demonstrate that TAID's performance improves relative to other methods as the task complexity increases. With its larger number of classes and more diverse images, ImageNet presents a more challenging scenario where TAID's adaptive interpolation mechanism shows more significant gains. This aligns with our observations in the main text that TAID's strengths are particularly evident in tasks with higher complexity and entropy.

## E MODEL DETAILS

### E.1 TAID-LLM-1.5B

For the development of `TAID-LLM-1.5B`, we utilized the full SmolLM-Corpus dataset. The training process consisted of 2 epochs, employing the AdamW optimizer with a cosine learning rate scheduler. We set the initial learning rate to $1\mathrm{e}{-5}$.

In this experiment, we used `Qwen2-72B-Instruct` as the teacher model and `Qwen2-1.5B-Instruct` as the student model. For the TAID-specific parameters, we used a momentum coefficient ($\beta$) of 0.99 and a learning rate of $t$ ($\alpha$) of $5\mathrm{e}{-5}$. The initial value of $t$ ($t_{\mathrm{start}}$) was set to 0.4, and the final value ($t_{\mathrm{end}}$) was set to 1.0.

To enhance training efficiency, we pre-computed the probabilities from the teacher model. Furthermore, to manage storage costs effectively, we only utilized the top 50 probabilities. This approach allowed us to balance computational resources and model performance, enabling efficient knowledge transfer from the large teacher model to the smaller student model.

Table 9 presents the detailed results for TAID-LLM-1.5B and other state-of-the-art small language models across various tasks as evaluated using the LightEval benchmark suite (Allal et al., 2024).

Table 9: **Performance of `TAID-LLM-1.5B`**, our new state-of-the-art LLM for models under 2B parameters.

| Model | MMLU | TriviaQA | ARC | PIQA | Hellaswag | OBQA | Winogrande | Average |
|---|---|---|---|---|---|---|---|---|
| Qwen2-1.5B (Yang et al., 2024) | 37.91 | 1.38 | 48.12 | 75.30 | 63.87 | 36.80 | 59.98 | 46.19 |
| Qwen2.5-1.5B (Qwen Team, 2024) | 41.15 | 0.68 | 58.41 | 76.01 | 66.40 | 40.00 | 59.35 | 48.86 |
| Phi-1.5B (Li et al., 2023a) | 35.92 | 6.06 | **60.53** | 75.62 | 60.72 | **46.00** | **67.88** | 50.39 |
| StableLM-2-1.6B (Bellagente et al., 2024) | 36.21 | **29.59** | 53.57 | 76.77 | 66.60 | 37.20 | 58.72 | 51.24 |
| SmolLM-1.7B (Allal et al., 2024) | **39.97** | 22.56 | 59.95 | 76.06 | 62.91 | 42.80 | 54.91 | 51.31 |
| **TAID-LLM-1.5B** | 39.96 | 22.96 | 58.14 | **77.37** | **67.15** | 41.40 | 58.88 | **52.27** |

Table 10: **Performance of `TAID-VLM-2B`**, our new state-of-the-art VLM for models up to 4B parameters.

| Model | MMBench_V11 | MMStar | MMMU_VAL | MathVista | OCRBench | AI2D | HallusionBench | MMVet | Average |
|---|---|---|---|---|---|---|---|---|---|
| PaliGemma-3B-mix-448 (Beyer et al., 2024) | 65.6 | 48.3 | 34.9 | 28.7 | 61.4 | 68.3 | 32.2 | 33.1 | 46.6 |
| MiniCPM-V-2 (Yao et al., 2024) | 65.8 | 39.1 | 38.2 | 39.8 | 60.5 | 62.9 | 36.1 | 41.0 | 47.9 |
| Phi-3-Vision (Abdin et al., 2024) | 65.2 | 47.7 | **46.1** | 44.6 | 63.7 | **78.4** | 39.0 | **44.1** | 53.6 |
| InternVL2-2B (Chen et al., 2024) | 69.6 | **49.8** | 36.3 | 46.0 | 78.1 | 74.1 | 38.0 | 39.7 | 54.0 |
| **TAID-VLM-2B** | **70.7** | 49.5 | 35.1 | **51.6** | **78.6** | 74.0 | **56.8** | 35.1 | **56.4** |

LightEval is designed to comprehensively assess the capabilities of small language models through a series of seven zero-shot tasks. Note that the scores in Table 4 denotes the average scores in Table 9.

As shown in Table 9, `TAID-LLM-1.5B` achieves competitive or superior performance across all tasks, with particularly strong results in PIQA and Hellaswag. This demonstrates the effectiveness of our distillation approach in creating a compact model that maintains high performance across a diverse range of language tasks.

### E.2 TAID-VLM-2B

For `TAID-VLM-2B`, we trained on the Mantis-Instruct dataset (Jiang et al., 2024). The training process spanned 3 epochs, using the AdamW optimizer with a cosine learning rate scheduler. The initial learning rate was set to $1e-6$.

In this vision-language model distillation task, we employed `InternVL2-8B` (Chen et al., 2024) as the teacher model and `InternVL2-2B` as the student model. The TAID-specific parameters remained largely consistent with those used for `TAID-LLM-1.5B`, with a momentum coefficient ($\beta$) of $0.99$ and $t_{start}$ of $0.4$. However, we adjusted the learning rate of $t$ to $5e-4$ to accommodate the characteristics of vision-language model training. The $t_{end}$ value was maintained at $1.0$.

Table 10 presents the detailed results for `TAID-VLM-2B` and other state-of-the-art small vision-language models across various tasks. Note that the scores in Table 5 denotes the average scores in Table 10.

As shown in Table 10, `TAID-VLM-2B` achieves competitive or superior performance across most tasks, with particularly strong results in MMStar, and HallusionBench. This demonstrates the effectiveness of our distillation approach in creating a compact vision-language model that maintains high performance across a diverse range of multimodal tasks.

