# OpenReview forum: "TAID: Temporally Adaptive Interpolated Distillation for Efficient Knowledge Transfer in Language Models"
_ICLR.cc/2025/Conference — ICLR 2025 Spotlight_

### Official Review · Reviewer_qQNq · 2024-10-24

**Soundness:** 3
**Presentation:** 3
**Contribution:** 2
**Rating:** 6
**Confidence:** 5

**Summary:**

This method addresses key challenges such as the capacity gap between teacher and student models and issues like mode averaging and mode collapse during distillation. The core idea behind TAID is the introduction of an intermediate, dynamic distribution that gradually shifts from the student model’s distribution to the teacher’s, enabling a smoother and more effective transfer of knowledge. The authors provide both theoretical and empirical analysis, showing that TAID prevents mode collapse, enhances performance across different model sizes, and results in more efficient and high-performing LLMs.

**Strengths:**

1. **Adaptive Approach**: The use of a dynamic, time-dependent intermediate teacher distribution addresses the capacity gap effectively, making it easier to transfer knowledge between models of different sizes.

2. **Versatility**: TAID is effective across a variety of model sizes and tasks, including language-only and vision-language models, making it broadly applicable.

3. **Theoretical Perspective**: TAID’s approach is supported by solid theoretical analysis, proving its ability to prevent mode collapse and balance between mode averaging and collapse, which traditional KD methods often struggle with. However, I didn't check the detailed proof for the theorem.

**Weaknesses:**

1. Despite the basic concept of the objective function being very similar to Skew KL in DistiLLM, there is limited comparison between Skew KL and TAID. I am wondering if there is a reasonable analysis of the main differences between the Skew KL and TAID functions.

**Questions:**

1. Does TAID also hold for reverse KL divergence? Are there any results regarding this?

---

> ### Author Response · Authors · 2024-11-18
>
> We sincerely thank you for your careful analysis of our technical approach, especially regarding the relationship between TAID and other distillation methods. Your questions have helped us clarify key distinctions and provide additional empirical evidence. We address each point in detail below.
>
> ### 1. Comparison to Skew KL
>
> > Despite the basic concept of the objective function being very similar to Skew KL in DistiLLM, there is limited comparison between Skew KL and TAID. I am wondering if there is a reasonable analysis of the main differences between the Skew KL and TAID functions.
>
> Thank you for raising this important question about the relationship between TAID and Skew KL. While they both leverage intermediate distributions, they have two fundamental differences:
>
> **1. Direction of Knowledge Flow:**
> We firstly summarize those objective functions:
> - TAID: $J_{\mathrm{TAID}} = \mathrm{KL}[p_t || q_\theta]$, where $p_t = (1-t)q + tp$
> - Skew KL: $J_{\mathrm{SKL}} = \mathrm{KL}[p || r]$, where $r = \lambda p + (1-\lambda)q_\theta$
>
> The position of interpolated distribution in KL divergence determines the knowledge flow:
> - TAID: $p_t → q_θ$ (intermediate distribution teaches student)
> - Skew KL: $p → r$ (teacher teaches intermediate distribution)
>
> **2. Static vs Dynamic Design:**
> - TAID uses a time-dependent parameter t that gradually changes during training
> - Skew KL uses a fixed interpolation parameter $\lambda$ throughout training
>
> Our empirical study (see table), following Section 6.3.2, validates the benefits of these design choices:
> - TAID shows consistent improvement as teacher size increases (410M → 6.9B)
> - Skew KL's performance degrades with larger teachers
>
> | Method | 410M  | 1B    | 2.8B  | 6.9B  |
> | :---: | ---: | ---: | ---: | ---: |
> | TAID   | 20.82 | 21.17 | 21.70 | 22.01 |
> | SKL    | 18.65 | 18.50 | 18.28 | 18.20 |
>
> The results demonstrate that TAID's direct knowledge transfer to student and dynamic nature effectively handles the capacity gap between teacher and student models.
>
> We will add this analysis to the final version to clarify these important distinctions.
>
> ### 2. TAID using Reverse KL
>
> > Does TAID also hold for reverse KL divergence? Are there any results regarding this?
>
> Yes, TAID can be extended to use reverse KL, and we have several theoretical and empirical insights to share:
>
> **1. Theoretical Extension:**
> TAID's objective function with reverse KL becomes:
> - TAID (KL): $J_{\mathrm{TAID}} = \mathrm{KL}[p_t || q_\theta]$
> - TAID (RKL): $J_{\mathrm{TAID}} = \mathrm{KL}[q_\theta || p_t]$
>
> **2. Empirical Results:**
> We conducted experiments for 2 teacher-student pairs using the same hyperparameters as in Section 6.1:
>
> | Method || MT-Bench Score |
> |:--:|--:|--:|
> | | Phi-3-mini → TinyLLaMA | StableLM → Pythia |
> | KL  | 2.71  | 2.74 |
> | RKL  | 3.48  | 2.53  |
> | TAID (KL)  | 4.05  | 3.05 |
> | TAID (RKL) | 3.36 | 2.54  |
>
> Interestingly, while TAID with standard KL shows significant improvement, TAID with RKL shows minimal changes from the baseline RKL. We hypothesize this is due to RKL's mode-seeking behavior: in the early stages of training, RKL may cause the student to overly commit to its initial modes, making it difficult to capture the teacher's modes even as t increases.
>
> This suggests that different objective functions may require different optimization strategies within the TAID framework. For example, with RKL, adjusting α to reach $p_t ≈ p$ earlier in training might be beneficial.
>
> Understanding which divergence measures work well (or poorly) with TAID's dynamic intermediate distribution, both theoretically and empirically, is an important direction for future research. We will explore these directions in future work to further advance our understanding of knowledge distillation with dynamic intermediate distributions.

---

> > ### Comment · Reviewer_qQNq · 2024-11-19
> >
> > I appreciate the feedback from the authors. I have carefully read the authors' response and the other reviewers' comments. I think my concerns are addressed and the paper became more convincing. As such I vote for accepting this paper.

---

### Official Review · Reviewer_jQRU · 2024-10-31

**Soundness:** 3
**Presentation:** 3
**Contribution:** 3
**Rating:** 8
**Confidence:** 5

**Summary:**

The paper focuses on improving the distillation procedure for LLMs and VLMs. It discusses the student-teacher distribution mismatch in standard distillation and identifies two crucial problems: the capacity gap and mode averaging/collapse. A novel approach based on time-dependent corrections to intermediate teacher distributions is proposed.
The method is compared to other techniques in the LLM setting, showing superior results. Finally, the paper presents results for distillation in both LLMs and VLMs.

**Strengths:**

- The problem formulation is well done. The paper discusses issues with current KD, identifies two specific problems, and addresses them effectively, creating a compelling narrative.
- The paper addresses both LLM and VLM distillation.
- The approach is straightforward, which is a strength. It requires adjusting only a single hyperparameter.
- The resulting models demonstrate state-of-the-art (SOTA) performance compared to models in the literature, as shown in Tables 4 and 5.

**Weaknesses:**

1. The results in Table 2 show that TAID performs very similarly to standard KL. They also indicate that other techniques underperform compared to KL. It is unclear if this is due to implementation.
2. The authors discuss the capacity gap and mode collapse in standard distillation. Is there any experimental evidence of this for LLM and VLM cases?
3. The paper claims to outperform all other distillation techniques. An analysis comparing the differences in loss functions between methods would be important to support this claim and to explain why the proposed approach is superior.
4. For Tables 4 and 5, it is important to assess improvements by comparing them to standard cross-entropy (CE) loss training.

**Questions:**

1. It is known that top-k logit distillation aids knowledge distillation (KD). Did the authors conduct experiments involving this? By top-k, I mean applying KD only on the top 100/1000 neurons to reduce the capacity gap.
2. The approach potentially bears similarities to linearly increasing the loss weight during distillation via t. Any thoughts on this?
3. What does m represent in line 199?
4. Were the results in Table 1 computed by the authors using the same data and settings?
5. How would this approach compare to adjusting the learning rate during the distillation process? Could it have a similar effect?
6. Will the code be open-sourced?
7. Is it beneficial to combine this with cross-entropy (CE) loss? If so, what should the ratio be?

---

> ### Author Response · Authors · 2024-11-19
>
> We sincerely thank you for your thorough evaluation and positive assessment of our work. Your detailed technical questions have helped us better articulate the strengths of our method. We have carefully addressed each point below.
>
> ### 1. Comparing to KL in Table 2
>
> > The results in Table 2 show that TAID performs very similarly to standard KL. They also indicate that other techniques underperform compared to KL. It is unclear if this is due to implementation.
>
> Thank you for this observation about Table 2. Let us clarify several important points:
> First, Table 2 shows results from the continual pre-training scenario, which is notably different from **the instruction tuning setting where most prior works focus**. Interestingly, a concurrent work [1], which specifically studied pre-training distillation, reported similar findings in Table 1 - standard KL consistently performs second-best.
>
> Therefore, we believe these results reflect fundamental characteristics of the pre-training task rather than implementation issues.
> It's known that pre-training is critical for knowledge acquisition while fine-tuning may increase hallucination [2], motivating our focus on pre-training distillation.
> One key factor might be the token distribution diversity:
> - Instruction data typically contains structured 2-person conversations, where pronouns like "I" and "you" dominate as subjects
> - Pre-training corpora include diverse subjects including proper nouns, resulting in flatter teacher distributions
> - This flatter distribution may explain why forward KL, which tends toward mode-averaging, performs well in pre-training scenarios
>
> [1] Gu et al., arXiv:2410.17215.
>
> [2] Gekhman et al., arXiv:2405.05904.
>
> ### 2. Evidence of Capacity Gap and Mode Collapse
>
> > The authors discuss the capacity gap and mode collapse in standard distillation. Is there any experimental evidence of this for LLM and VLM cases?
>
> Thank you for this important question. We have conducted comprehensive analyses for LLM in Section 6.3, and would appreciate any specific aspects you'd like us to investigate further.
>
> **Capacity gap:**
> Figure 2 (Right) empirically demonstrates this issue by comparing different teacher sizes. While TAID shows monotonic improvement with increasing teacher size, conventional methods exhibit inconsistent or degraded performance, clearly indicating their difficulty in handling large capacity gaps.
>
> **Mode collapse:**
> Table 3 quantitatively demonstrates mode collapse for RKL through probability mass distribution analysis: RKL shows clear signs of mode collapse with high head probability.
>
> **VLM case:**
> To our knowledge, there's no study specifically analyzing these issues for VLM. So, investigating these phenomena in VLM distillation remains an interesting direction for future research, as it requires careful consideration of the unique characteristics of vision-language representations.
>
> ### 3. Loss Comparison
>
> > An analysis comparing the differences in loss functions between methods would be important to support this claim and to explain why the proposed approach is superior.
>
> We would like to clarify that TAID proposes a new loss function itself (Eq. 2) based on the intermediate distribution (Eq. 1).
>
> Our experimental results consistently demonstrate the effectiveness of our loss function across different scenarios:
> - **1. Instruction tuning (Table 1):** Most methods are pure loss function variants, except for GKD and DistiLLM which use student-generated outputs (SGOs). TAID outperforms all these methods.
> - **2. Pre-training (Table 2):** Different loss functions are compared using the same training setup, excluding additional techniques like SGOs. TAID achieves the best performance in this comparison too.
>
> If you have specific aspects of the loss function comparison that you would like us to elaborate on, we would be happy to provide additional analysis.
>
> ### 4. Top-k Logit Distillation
>
> > It is known that top-k logit distillation aids KD. Did the authors conduct experiments involving this?
>
> We appreciate your suggestion about top-k logit distillation. While we could not find specific references about “it is known that top-k logit distillation aids knowledge distillation", we are interested in understanding more about this approach.
>
> To investigate this, we conducted additional experiments as Section 6.3.2, specifically focusing on the largest capacity gap case (Pythia 6.9B → 70M):
>
> | Method | Test Accuracy |
> |:--:|--:|
> | TAID | **22.01** |
> | TAID (top-k=100)  | 19.91 |
>
> These results suggest that, at least in TAID, top-k distillation does not help bridge the capacity gap. We hypothesize this might be because limiting distillation to top-k logits loses the rich distributional information that makes knowledge distillation effective.
> We would be interested in exploring this direction further with additional references you might suggest.
>
> ------
> Due to the character limit, we will address the remaining points in our second response.

---

> ### Author Response · Authors · 2024-11-19
>
> This is a continuation of our previous response. We address the remaining questions below.
>
> -----
>
> ### 5. Comparison to Linearly Weighted KL
>
> > The approach potentially bears similarities to linearly increasing the loss weight during distillation via t. Any thoughts on this?
>
> Thank you for this insightful observation. While these approaches share formal similarities in their use of parameter $t$, TAID's interpolation fundamentally differs in that it operates in the logit space (Line 160), directly modifying the target distribution that the student tries to match, rather than just modulating the strength of the learning signal. This difference in the mathematical formulation leads to distinct gradient behavior during training.
>
> We empirically verified the importance of this distinction by comparing TAID with linearly weighted KL distillation using the same experimental setup as Section 6.1. The results on MT-Bench show:
>
> | Method | Score (Phi-3-mini → TinyLlama) |
> |:--:|--:|
> | Linearly weighted KL   | 2.45 |
> | TAID w/o adaptive update | **3.48** |
>
> This significant performance gap demonstrates that TAID's interpolation mechanism offers substantial benefits beyond what can be achieved through simple loss weighting.
>
> ### 6. About m in Line 199
>
> > What does m represent in line 199?
>
> The m in line 199 represents the momentum term in our adaptive update mechanism. Specifically, the momentum at training step $n$ is computed by:
> $$ m_n = \beta m_{n-1} + (1-\beta) \delta_n$$,
> where $beta$ is the momentum coefficient and $\delta_n$ is the relative change in the objective function. It is used to update the interpolation parameter $t_n$. This momentum-based approach helps smooth out short-term fluctuations in the interpolation parameter updates, leading to more stable training.
>
> We will revise the manuscript to make this definition and its role clearer in the camera-ready version.
>
> ### 7. Experimental Setting in Table 1
>
> > Were the results in Table 1 computed by the authors using the same data and settings?
>
> > Will the code be open-sourced?
>
> Yes, all results in Table 1 were computed by us under identical experimental settings to ensure fair comparison. For baseline implementations, we used the official code from DistiLLM [3], which has been publicly released.
>
> Our complete implementation is included in the supplementary materials and will be open-sourced soon.
>
> [3] Ko et al., arXiv:2402.03898.
>
> ### 8. Comparison to Adjusting Learning Late
>
> > How would this approach compare to adjusting the learning rate during the distillation process? Could it have a similar effect?
>
> This is an insightful question about optimization dynamics. While adjusting the learning rate (LR) might appear similar to TAID's interpolation parameter $t$, they operate fundamentally differently:
>
> **1. Mathematical Comparison:**
> TAID's interpolation (Eq. 1) influences the gradient itself by modifying the target distribution, while LR scaling only affects the magnitude of the gradient step:
> - LR adjustment: $\theta ← \theta - \eta(t)\nabla \mathrm{KL}(p||q_\theta) $
> - TAID: $\theta ← \theta - \eta \nabla \mathrm{KL}(p_t||q_θ)$
>
> **2. Impact on Optimization:**
> - LR adjustment: Simply scales how much we move towards a fixed target
> - TAID: Directly shapes the target distribution, allowing for more controlled knowledge transfer
>
> **3. Complementary Nature:**
> TAID can be effectively combined with learning rate scheduling to ensure stable and effective knowledge transfer.
>
> Based on our analysis of linearly weighted KL in the previous response (5. Comparison to Linearly Weighted KL), we expect learning rate adjustment alone would show similar limitations, while TAID provides more fine-grained control over the knowledge transfer process.
>
> ### 9. Combination with CE loss
>
> > Is it beneficial to combine this with cross-entropy (CE) loss? If so, what should the ratio be?
>
> In our main experiments, we intentionally excluded the CE loss to isolate and compare the pure effect of different distillation loss functions. At the same time, your question raises an important practical consideration. In our model development experiments (Section 6 and the other internal experiments), we have observed that adding CE loss can further improve performance. As evidence, the below table demonstrates the effectiveness:
>
> | Method | Score (StableLM → Pythia) |
> | :--: | --: |
> | TAID | 3.05 |
> | TAID + CE (ratio = 0.2) | **3.07** |
>
> The optimal CE ratio appears to be model-dependent, with larger vocabulary models (e.g., Qwen2) benefiting from higher ratios (0.5-0.8).
> We believe that systematic investigation of these relationships - particularly the interaction between vocabulary size, model architecture, and optimal loss ratios - is crucial for developing more efficient and high-performing compact models. This is an important direction for future work that could further advance the development of practical, resource-efficient language models.

---

> > ### Comment · Reviewer_jQRU · 2024-11-23
> > **Acknowledgment of rebuttal**
> >
> > Thank you for providing a constructive rebuttal.
> >
> > Additional question after reading the response:
> > 1) For point 5. authors mention "his difference in the mathematical formulation leads to distinct gradient behavior during training." I would like to see mathematical derivation of gradients for these 2 cases.

---

> > > ### Author Response · Authors · 2024-11-25
> > > **Thank you for your response and further question!**
> > >
> > > For linearly weighted KL $(1-t) \mathrm{KL}(q’ \parallel q_\theta) + t \mathrm{KL}(p \parallel q_\theta)$, the gradient is:
> > >
> > > $\nabla_\theta t \mathrm{KL}(p \parallel q_\theta) = t \cdot \left(- E_p [ \nabla_\theta \log q_\theta(y) ] \right)$.
> > >
> > > For TAID, our gradient involves the interpolated distribution $p_t$:
> > >
> > > $\nabla_\theta \mathrm{KL}(p_t \parallel q_\theta) = \nabla_\theta E_{p_t}[- \log q_\theta(y)] = -E_{p_t}[\nabla_\theta \log q_\theta(y)].$
> > >
> > > Therefore, the two formulas are fundamentally different when $p_t$ is defined through logit-space interpolation (as we detail subsequently soon); although they are equivalent up to a constant when $p_t$ is defined as the probability-space interpolation $p_t=(1-t)q’+tp$.
> > >
> > > As mentioned in Line 160 in the paper and implemented in Line 72 of `src/distil_losses/taid.py` in our supplementary material, we perform interpolation at the logit level. We found that Eq. (1) in the paper incorrectly presents interpolation at the probability level. Thank you for bringing this to our attention as it helps improve the clarity of our presentation.
> > >
> > > The correct logit-space formulation of TAID interpolated distribution $p_t$ should be:
> > >
> > > $p_t(y_s|y^{<s}) := \mathrm{softmax}\left((1-t) \cdot \mathrm{logit}_{q'}(y_s | y^{<s}) + t \cdot  \mathrm{logit}_p (y_s | y^{<s})  \right)$,
> > >
> > > where $t \in [0, 1]$ is a time-dependent interpolation parameter, $\mathrm{logit}_{q'}$ represents a detached version of the student logits (i.e., treated as a constant without being backpropagated), and $\mathrm{logit}_p$ represents the teacher logits.
> > > This logit-space interpolation preserves the relative strengths of predictions before normalization, allowing the student to learn from richer distributional information.
> > >
> > > As shown in our previous response, TAID leads to significantly better performance compared to linearly weighted KL, and the adaptive update mechanism further enhances this advantage by dynamically adjusting the interpolation strength.
> > >
> > > We have updated Eq. (1) in the revised manuscript to show the correct logit-space formulation. We sincerely appreciate your careful review, which helped us identify this discrepancy between the formulation and implementation, leading to a more accurate presentation of our method.

---

> > > > ### Author Response · Authors · 2024-12-02
> > > > **Follow-up on gradient derivation**
> > > >
> > > > We would like to thank reviewer jQRU once again for your careful review, particularly your question about gradient derivation which helped us improve the mathematical precision of our work. We have already updated the manuscript accordingly.
> > > >
> > > > We hope our response has clearly demonstrated the distinction between TAID and linearly weighted KL. As the discussion period is coming to a close, we would greatly appreciate if you could let us know whether you found our explanation satisfactory.
> > > >
> > > > Please do not hesitate to raise any further questions about our technical analysis.

---

### Official Review · Reviewer_RYft · 2024-11-02

**Soundness:** 3
**Presentation:** 3
**Contribution:** 3
**Rating:** 8
**Confidence:** 5

**Summary:**

This paper proposes TAID, introducing a temporally adaptive interpolation between the student and teacher distributions. The interpolation is upon relative change in the objective function. Numerical experiments over instruction tuning and pretraining stage well demonstrated the efficacy of TAID.

**Strengths:**

- The paper is well motivated, written well and easy to follow.

- The numerical experiments are comprehensive to well show the effectiveness of TAID.

**Weaknesses:**

- The technical contribution feels somewhat limited, primarily focusing on the difference between TAID and Skew KLD through the proposed temporal adaptive interpolation mechanism. This design choice, however, appears heuristic in nature. Given that multiple adaptive mechanisms could potentially meet the provided criteria, it would be helpful for the authors to elaborate on why this particular format was chosen for the adaptive mechanism. Providing additional rationale or theoretical support would strengthen the justification for this approach and enhance the perceived novelty of the contribution.

**Questions:**

See the weakness.

---

> ### Author Response · Authors · 2024-11-18
>
> We sincerely thank you for your thorough and constructive feedback. Your thoughtful question have helped us identify important areas for clarification. We have carefully addressed your question below and will include in the camera-ready version.
>
> ### About TAID’s Adaptive Mechanism
>
> > Given that multiple adaptive mechanisms could potentially meet the provided criteria, it would be helpful for the authors to elaborate on why this particular format was chosen for the adaptive mechanism. Providing additional rationale or theoretical support would strengthen the justification for this approach and enhance the perceived novelty of the contribution.
>
> Thank you for this important question about our adaptive mechanism design. Our mechanism was designed to achieve robust performance across different models without extensive hyperparameter tuning. In the camera-ready version, we will provide a detailed discussion of the advantages of our proposed method described below.
>
> **1. Key Design Requirements and Solutions:**
> Our adaptive mechanism satisfies three critical requirements through specific design elements:
>
> - a) **Scale Invariance:** Ensure applicability across model sizes
>   - → Achieved by relative loss measure (Line 10 in Algorithm 1)
> - b) **Stability:** Prevent abrupt changes in the intermediate distribution
>   - → Achieved by momentum term (Line 11 in Algorithm 1)
> - c) **Boundedness:** Guarantee $t \in [0,1]$
>   - → Achieved by sigmoid activation in the update rule (Line 12 in Algorithm 1)
>
> **2. Alternative Approaches and Analysis:**
> We explored several potential design alternatives including the following ones before arriving at our current approach:
>
> - **Linear scheduling:** A straightforward approach that increases t linearly over training steps. While simple to implement, it cannot adapt to the student's learning progress (Figure 2 (Left)). This is our motivation for the adaptive update.
>
> - **Direct loss-based update:** Similar to the update rule in Group DRO [1], where $t$ is updated based on the loss value directly (not relative loss). This approach has two key issues: (1) Loss scale sensitivity: Requires different hyperparameter tuning for different model sizes and types (2) Update behavior: Tends to make abrupt changes based on absolute loss values.
>   To analyze the update behavior, we conducted an empirical comparison between direct loss (natural alternative)  and relative loss (our approach) updates. The result (see [figure](https://drive.google.com/file/d/15YYxLjSAWeoEcFbpYi0eBSLFSC8onkgX/view?usp=sharing), URL is anonymized) demonstrates the second issue clearly: direct loss updates lead to abrupt early changes and premature convergence, while our relative loss approach enables smooth, continuous adaptation regardless of the absolute loss scale.
>
> - **Update without momentum:** A simplified version of our approach that removes the momentum term. This leads to potential instability in updates. Our analysis (see [figure](https://drive.google.com/file/d/1rZLcPBRgtnr-9FObtn49TeJKMYbbJa1c/view?usp=sharing), URL is anonymized), showing the evolution of the interpolation parameter t during the initial training steps, reveals that without momentum exhibits unstable updates, while TAID with momentum maintains smooth and stable progression.
>
>
> After extensive experimentation with these alternatives, we found our adaptive approach with relative loss measure consistently performed best across different settings.
>
> **3. Future Directions:**
> Exploring improved adaptive mechanisms while maintaining the current benefits of stability and scale invariance is an interesting direction for future work. This point will also be explicitly addressed in the camera-ready version.
>
> We sincerely appreciate your thoughtful feedback on the critical issue of justifying the proposed method.
>
> [1] Sagawa et al., Distributionally Robust Neural Networks for Group Shifts: On the Importance of Regularization for Worst-Case Generalization, 2019.

---

> > ### Author Response · Authors · 2024-12-02
> > **Follow-up to Rebuttal**
> >
> > We would like to thank reviewer RYft once again for your time and effort in reviewing our work.
> >
> > We hope our response has helped clarify your questions about TAID's adaptive mechanism design. As the discussion period is coming to a close, we would greatly appreciate if you could let us know whether you found our response satisfactory and share your thoughts.
> >
> > Please do not hesitate to raise any further questions or concerns about our work.

---

> > > ### Comment · Reviewer_RYft · 2024-12-02
> > >
> > > I thank the authors for the detailed response. I decide to increase my score to 8.

---

### Meta-Review · Area_Chair_NAPg · 2024-12-18

**Metareview:**

The paper proposes a novel distillation technique for LLMs that mitigates issues in common distillation techniques such as mode collapse. Their TAID approach dynamically interpolates the training signal to the distribution between the student and to the teacher.
The paper is well motivated, written well and easy to follow. The numerical experiments are comprehensive and show the effectiveness of TAID.

**Additional Comments On Reviewer Discussion:**

Some initial concerns about the technical contribution were raised but were resolved in the rebuttal, leading to an increase in score from the reviewers.

---

### Decision · Program_Chairs · 2025-01-22

Accept (Spotlight)